# HumanoidGen: Data Generation for Bimanual Dexterous Manipulation via LLM Reasoning

Zhi Jing[1,2]    Siyuan Yang[3,2]    Jicong Ao[2]    Ting Xiao[4]    Yu-Gang Jiang[1]    Chenjia Bai[† 2]

[1]Fudan University[‡]    [2]Institute of Artificial Intelligence (TeleAI), China Telecom[‡]
[3]University of Science and Technology of China
[4]East China University of Science and Technology

## Abstract

For robotic manipulation, existing robotics datasets and simulation benchmarks predominantly cater to robot-arm platforms. However, for humanoid robots equipped with dual arms and dexterous hands, simulation tasks and high-quality demonstrations are notably lacking. Bimanual dexterous manipulation is inherently more complex, as it requires coordinated arm movements and hand operations, making autonomous data collection challenging. This paper presents HumanoidGen, an automated task creation and demonstration collection framework that leverages atomic dexterous operations and LLM reasoning to generate relational constraints. Specifically, we provide spatial annotations for both assets and dexterous hands based on the atomic operations, and perform an LLM planner to generate a chain of actionable spatial constraints for arm movements based on object affordances and scenes. To further improve planning ability, we employ a variant of Monte Carlo tree search to enhance LLM reasoning for long-horizon tasks and insufficient annotation. In experiments, we create a novel benchmark with augmented scenarios to evaluate the quality of the collected data. The results show that the performance of the 2D and 3D diffusion policies can scale with the generated dataset. Project page is https://openhumanoidgen.github.io.

## 1 Introduction

The long-term goal of embodied manipulation is to achieve human-like manipulation capabilities across versatile scenes and tasks [1, 2]. Humanoid robots, with their human-like morphology, offer a universal platform capable of leveraging bimanual coordination and dexterous hand manipulation [3, 4], potentially enabling more complex tasks than conventional robot arms. Bimanual dexterous manipulation is inherently intricate due to the requirement for coordinated arm movements and hand operations, which in turn makes the collection of the demonstration dataset more challenging.

The existing data generation pipeline for humanoid robots primarily depends on teleoperation systems via Virtual Reality (VR) [5, 6] and exoskeleton devices [7, 8, 9]. However, teleoperation requires the deployment of numerous real-world objects and scenes, and also requires proficient operational skills from human operators. This makes it difficult for the dataset to cover diverse real-world tasks and scenarios. Consequently, existing real-world datasets such as Open-X [10, 11, 12, 13] consist mainly of single-arm manipulation data. To address this issue, many researchers have turned to various simulators for data acquisition, such as SAPIEN [14], IsaacSim [15], and MuJoCo [16]. In the context of bimanual manipulation, simulated benchmarks such as Aloha [17], PerAct2 [18], and RoboTwin [19] utilize dual-arm or wheeled robots for task creation and data collection. These efforts are limited

---

[†] Correspondence to: Chenjia Bai (baicj@chinatelecom.cn)    [‡] Equally leading organizations

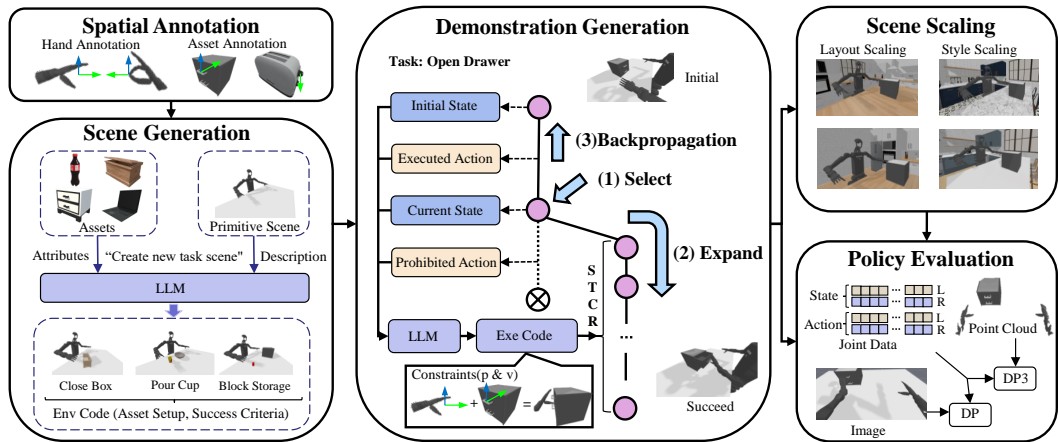

Figure 1: The overview of HumanoidGen. It includes spatial annotations, scene generation, constraint generation, MCTS-enhanced reasoning, data collection, scene scaling, and policy evaluation.

in terms of task diversity and generally do not involve dexterous hands. Similarly, HumanoidBench [20] and BiGym [21] use humanoid robots to perform loco-manipulation tasks but depend on VR or reinforcement learning (RL) policies for data collection, which would be costly considering that humanoid robots require coordinated control of arms and dexterous hands. Therefore, a more efficient approach is to allocate rich interactable assets in simulation for task creation and develop a fully autonomous paradigm for data collection. Taking inspiration from LLM-driven planning methods [22, 23], our goal is to generate code-form planning to complete humanoid manipulation tasks, aiming to produce high-quality demonstrations that can scale efficiently.

In this paper, we propose **HumanoidGen**, an LLM-based framework capable of generating diverse manipulation tasks and collecting scalable demonstrations for humanoid robots. For task creation, an LLM planner is prompted to generate code for environment setup and success criteria based on language descriptions and the wealth of 3D assets. To gather demonstrations for each task, we first identify atomic operations for dexterous hands (such as pinch and grab), then we adopt LLMs to perform task decomposition for long-horizon tasks, generating spatial rational constraints for arm movements. Specifically, we give spatial annotations for assets, and the constraints are defined through geometrical relationships on contact points and the function axis for both the arm and entities. Then we conduct LLM-based code generation by translating planning into executable code-form constraints. An off-the-shelf trajectory optimizer is applied subsequently to solve the constraints to get the arm and hand movements. To further enhance reasoning efficiency in complex tasks with long constraint chains, we incorporate a variant of Monte Carlo tree search (MCTS) for better test-time reasoning. This approach strengthens the reasoning ability of LLMs, especially when the required spatial annotations are absent.

Building on HumanoidGen, we develop a comprehensive benchmark called **HGen-Bench** for bimanual dexterous manipulation. In our setup, the Unitree H1-2 humanoid robot equipped with Inspire hands serves as the robotic platform, and SAPIEN [14] as the simulation engine for data collection. HGen-Bench consists of 20 tasks of varying difficulty levels. For each task, a chain of constraints that determines arm movements and hand operations is generated, taking into account the spatial relationship between the robot and the objects. Then the actions are obtained by solving the constrained optimization problem. We execute these actions in the simulator and record the successful trajectories as demonstrations. We evaluated the success rate in trajectory generation with randomized scene configurations. The results show that MCTS significantly improves the reasoning ability of LLMs for long-horizon tasks and insufficient annotations. We train both 2D and 3D diffusion policies on the generated data, and the results show that the performance of the policy improved continuously as more demonstrations are incorporated.

## 2 Method

HumanoidGen is an automated framework for scene generation, demonstration collection, and data generalization for bimanual dexterous manipulation, aiming to provide high-quality demonstrations over diverse scenarios to facilitate data scaling and policy learning. The overview of our framework

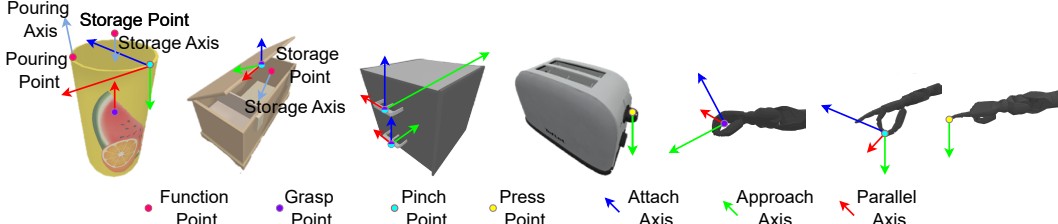

Figure 2: The spatial annotations, including key points and key axes for assets and hands, as well as the atomic operations of hands that include grasp, pinch, and press.

is shown in Fig. 1. (i) As preparation, the assets and dexterous hands are meticulously annotated with spatial information, allowing flexible and reliable LLM-based planning. In scene generation, the LLM planner aims to generate an environment setup with code-form configuration based on asset, scene, and task descriptions. (ii) Based on the generated scenes and pre-defined hand atomic operations, the LLM proceeds to generate the planning code of a chain of spatial constraints for subsequent data collection. (iii) For tasks with long-horizon planning and insufficient annotations, we employ MCTS with introspective exploration to enhance the reasoning ability of LLMs. (iv) Then, we collect demonstrations by executing the planning with augmented scenarios to enhance data diversity. These demonstrations are utilized to construct a humanoid manipulation benchmark for policy evaluation.

## 2.1 Spatial Annotation and Scene Generation

**Spatial Annotation Preparation.** For efficient LLM planning with atomic hand operations, we conduct meticulous annotations, including key points and key axes for both assets and dexterous hands. As shown in Fig. 2, we categorize annotations into three types as follows.

*Hand Atomic Operation.* The annotations of atomic operations indicate how the dexterous hands can interact with assets. For smooth operation execution, we perform the annotation for each atomic operation on both dexterous hands. As shown in Fig. 2, the key point and the axis vary in different atomic operations. When annotating the operation axes, we specifically categorized them into three types according to the finger movement of the atomic operation: *approach axes, attach axes*, and *parallel axes*. Take 'pinch' as an example, (i) its approach axis specifies the direction from which the dexterous hand should approach its pre-pinching pose. This ensures proper alignment for the pinching operation execution. (ii) The attach axis indicates the finger movement direction during the operation execution, pointing from the thumb tip to the index finger tip for 'pinch'. (iii) The parallel axis, perpendicular to the first two and oriented parallel to the palm plane, typically specifies the object's rotation axis during manipulation. This helps specify the rotation axis when manipulating an articulated object.

*Asset Inherent Information.* Asset inherent information indicates how the asset performs its own functionality. Usually, they are the points and axes where the asset exerts its functions, which are used to indicate the interaction between objects. These points and axes vary according to the functionality of the asset. For example, in Fig. 2, the cup has two possible functions: pouring and storage, which correspond to different annotations on the point and the axis, respectively.

*Asset Operation Annotations.* Asset operation annotations define the points and axes on an asset that indicate how various atomic operations can be applied to interact with it. Importantly, atomic operations do not have a one-to-one correspondence with assets: (i) Different operations can be performed on the same asset. For instance, both grasping and pinching can apply to a cup. (ii) The same operation can use different key points on the same asset. As shown in Fig. 2, grasping can target keypoints on both the upper and lower parts of a drawer. (iii) A single keypoint may support multiple operations, such as the endpoint of a box lid, which allows both grasping and pinching (Fig. 2).

With annotations, we can abstract the scene information into a set of key points and key axes in the local frames of objects and hands separately. This facilitates the LLM's spatial and relational understanding of the scene and tasks, providing the foundation for defining the constraints of various atomic operations. We also note that the recent approach [19] adopts Stable Diffusion to simplify the annotation process for similar assets, which can be integrated with our method. Implementation details and evaluation results are provided in Appendix A.1 and Appendix C.2.

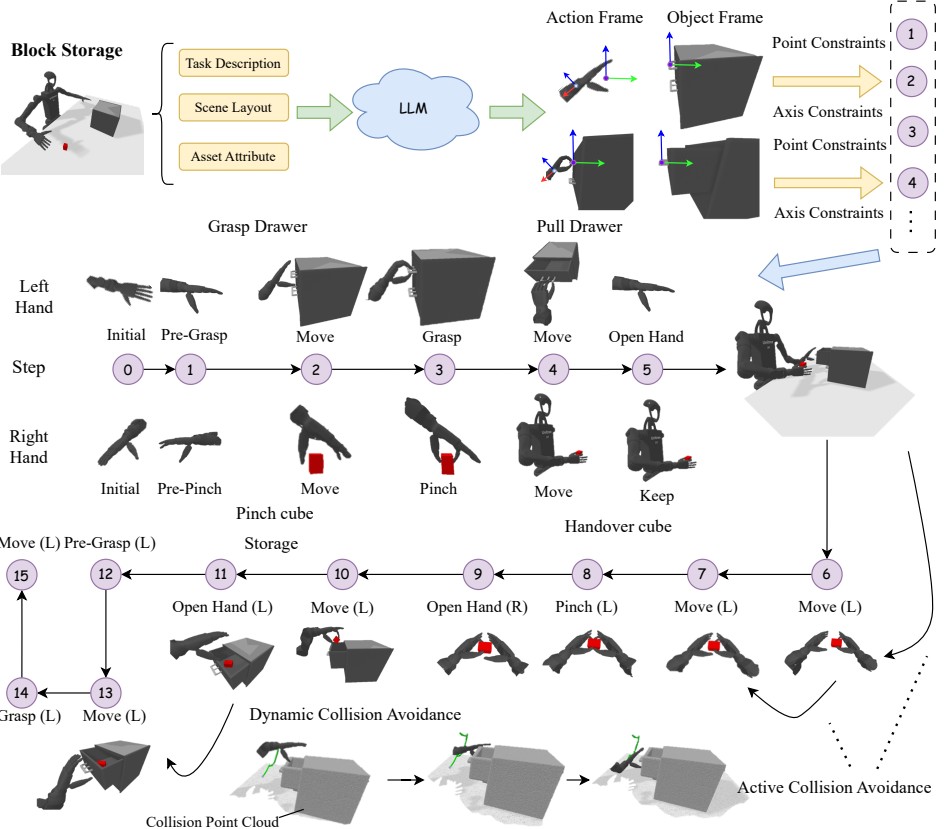

Figure 3: An illustration of the generated plan for the task *block storage*. The LLM is prompted with a task description, scene layout, and asset attributes to generate a step sequence. Each step is expressed using an atomic operation, along with its corresponding annotations. During plan execution, (i) from step 0 - 5, the left hand pulls out the drawer by grasping its handle, while the right hand simultaneously pinches and lifts the cube. (ii) From step 6 - 9, the left hand takes the cube from the right hand. (iii) From step 10 - 15, the left hand places the cube into the drawer and pushes the drawer back by grasping its handle. The LLM avoids collisions that would occur from directly moving to the pinching pose by planning a collision-free method during steps 6 - 7, demonstrating active collision avoidance. Additionally, the LLM proactively generates code to account for potential collisions with the drawer when performing free motion in the compact workspace, as illustrated in the bottom part.

**Scene Generation.** Based on the annotations, an LLM planner is prompted to generate tabletop-level task scenes and configurations according to the task description. To achieve this, we take the asset library information, scene details, and task requirements as the prompt, then the LLM planner generates task-setup code, determining the categories, quantities, placement poses, and other attributes of the selected assets. In addition, we employ LLMs to generate task success criteria to guide the execution process and determine when the task is completed. For instance, in the *cup pour* task, the LLM must infer the size of cubes contained in the cup based on the cup's size, and assign plausible positions for the cup, the cubes, and the target bowl. The success criterion is defined as all cubes being successfully placed inside the bowl.

## 2.2 LLM-based Task Planning

We design an automated generation method to generate scripts of constraint chains, which are used to collect demonstrations with the help of trajectory optimizers. Our method significantly differs from the conventional methods, which use LLMs only for high-level task plan generation and require a wide array of predefined low-level operations [24]. Specifically, our method includes task decomposition, relational action constraints, and collision avoidance to enhance planning abilities.

**Task Decomposition.** As shown in Fig. 3, the LLM planner decomposes the long-horizon task into an action sequence $\mathcal{S} = \{S_1, S_2, ..., S_n\}$ with $n$ steps according to the task description, where

each step $S_i = (S_i^l, S_i^r)$ includes left and right parts, with $S_i^l, S_i^r \in \{A_i, M_i\}$, where $A_i$ and $M_i$ indicate hand operations and arm movements respectively. (i) For hand operation, the LLM selects $A_i$ from the atomic library $\mathcal{A}^{\text{hand}} = \{A^{\text{pinch}}, A^{\text{grasp}}, ..., A^{\text{open}}\}$ at each step. (ii) For arm movements, we define two types of constraint: the goal pose constraint $C_i^{\text{goal}}$ and the path constraint $C_i^{\text{path}}$. Using these two constraints, the motion planning problem can be transformed into a constrained optimization problem, denoted as $M_i = \{f(C_i^{\text{goal}}), f(C_i^{\text{path}})\}$, where the constraint $C_i$ is inferred by the LLM based on contextual atomic operations and functional reasoning related to movement, and $f(\cdot)$ is a constraints solver detailed in Appendix A.2. By employing task decomposition, the long-horizon task can be decomposed into atomic operations and constraint definition, effectively reducing the reasoning complexity while enhancing planning efficiency.

**Relational Action Constraints.** Compared with conventional grippers, dexterous hands require more sophisticated motion constraints to accommodate diverse hand gestures and contact interactions. To systematically describe these relationships, we introduce a set of dynamic coordinate frames $\mathcal{F}_{\text{act}}$ that represent the contextual action space of each hand. Each frame $F = (p_1, \ldots, p_k; v_1, \ldots, v_m)$ in $\mathcal{F}_{\text{act}}$ consists of key points $p_i \in \mathbb{R}^3$ and axes $v_j \in \mathbb{R}^3$. The composition of $\mathcal{F}_{\text{act}}$ evolves dynamically according to the manipulation state: for instance, the left-hand action frame $\mathcal{F}_{\text{act}}^l$ initially includes only $\{F_l\}$, which defines the geometric properties of the hand itself, and extends to $\{F_l, F_o\}$ once a rigid grasp with object $o$ is established. Each constraint $c \in \{C^{\text{path}}, C^{\text{goal}}\}$ specifies a geometric relation between an element of $\mathcal{F}_{\text{act}}$ and one from the global frame set $\mathcal{F}_{\text{all}} = \mathcal{F}_{\text{obj}} \cup \mathcal{F}_{\text{hand}} \cup \mathcal{F}_{\text{world}}$. These relations can take the form of point or axis correspondences, such as coincidence, parallelism, or orthogonality, allowing flexible encoding of motion intents. For example, in a grasping action (step 2 in Fig. 3), a goal constraint may enforce point coincidence and directional alignment, $p_{\text{grasp}}^{\text{hand}} \equiv p_{\text{grasp}}^{\text{handle}}$ and $v_{\text{approach}}^{\text{hand}} \parallel v_{\text{approach}}^{\text{handle}}$. During the subsequent pulling phase (step 3), a path constraint $C^{\text{path}} = \{v_{\text{grasp}}^{\text{hand}} \parallel v_{\text{grasp}}^{\text{handle}}\}$ is applied to ensure that the hand maintains a stable gesture throughout the motion. This formulation explicitly encodes the spatial logic underlying dexterous manipulation, enabling consistent and adaptive control across complex, multi-stage tasks.

**Collision Avoidance.** Ensuring effective collision avoidance during atomic operations is a key challenge. We propose two solutions: (i) *Active Collision Avoidance.* The LLM planner generates atomic operation scripts that proactively incorporate collision avoidance behavior. This ensures the feasibility of the atomic operations, especially when a coordinate bimanual manipulation is necessary. For example, in *block stack*, the left hand retracts from the stacking area to make space for the right hand. (ii) *Dynamic Collision Management.* The LLM planner dynamically manages collision checks to enable in-contact manipulation. Specifically, we maintain an object-ignoring list and let the LLM dynamically adjust it when generating scripts, thereby providing guidance for successful low-level trajectory optimization and execution. This is crucial for in-contact articulated object manipulation, whereas existing methods ignore this aspect or rely on manual processing in atomic operation programming. For example, in the *block storage* task illustrated in Fig. 3, the LLM adds the drawer to the list to ignore the contact between the robot hand and the drawer when pulling the drawer out. After that, the drawer is removed from the list, enabling subsequent collision avoidance. This approach seamlessly integrates in-contact manipulation with free-space motion, significantly enhancing the framework's ability to handle complex long-horizon dexterous tasks.

## 2.3 Enhancing Reasoning with MCTS

When performing long-horizon tasks or encountering insufficiently annotated objects, LLMs lack sufficient prompts for reliable reasoning, which often leads to failed planning. To address this issue, we employ tree search to enhance the ability of multi-step reasoning in task planning. Specifically, we propose a novel *Segment-Truncate-Combine-Resume* (STCR) mechanism that abstracts a planning search tree from LLMs' outputs. By integrating MCTS exploration and exploitation strategies, the tree is iteratively expanded to derive an executable solution.

**STCR mechanism.** Inspired by abstractions in proving problems [25, 26], STCR aims to abstract executable code into a tree to define nodes and branches, which includes four steps. (i) ***Segment.*** By matching critical functions, the planning code is segmented into multiple steps according to the granularity of the execution. The segmentation is inferred by LLMs following a similar granularity as described in the task decomposition of §2.2, denoted as $\mathcal{S} = \{S_1, S_2, ..., S_n\}$. (ii) ***Truncate.*** The truncation is performed at the point where an error occurs. We execute the segmented steps sequentially until a code formatting error or action execution failure occurs (e.g., move or grasp

failed). We truncate the steps at the point of failure, discard the steps after the erroneous step, and retain the valid segments as $\mathcal{S}_{\mathrm{remain}} = \{S_i | 1 \leq i \leq k-1\}$, where $S_k$ failed. (iii) **Combine.** We merge atomic operations with consistent intent in $\mathcal{S}_{\mathrm{remain}}$ to form a new execution sequence $\mathcal{S}'_{\mathrm{remain}} = \{S'_1, S'_2, \ldots, S'_{k'}\}$, where $k' < k-1$. Operations such as *grasping* and *pinching* are implemented by multiple steps. These steps are abstracted into single execution units during tree search. For instance, if $S'_1$ represents a grasping action containing two move steps, then $S'_1 = \{A_1^{\mathrm{pre\_grasp}}, M_2, M_3, A_4^{\mathrm{grasp}}\}$ denotes the combined action sequence. In the task tree, each combined action sequence $S'$ represents an executable branch, and its terminal state serves as a new node in the tree structure. (iv) **Resume.** For newly created nodes, the code of the executed sequence, the error code, and the scene state information are stored, which will be resumed when this node is selected in expansion.

**Interactive Task Planning via MCTS.** Based on the tree built by STCR, we adopt MCTS as the tree search algorithm, consisting *Selection, Expansion*, and *Backpropagation* steps. Different from standard MCTS methods, we incorporate simulation steps into the expansion process to provide execution results of the nodes. The details are given in Appendix A.4.

**Scene Scaling.** To enhance the data diversity, we perform room-level scene scaling that allows demonstrations to contain diverse scenes using demonstration scripts generated from table-level task scenes. We compute the transformation matrix between the coordinate systems of the original and new scenes to align the two scenes. The details are given in Appendix A.3. By harnessing diverse assets in RoboCasa [24], we can generalize demonstrations across over 120 scenes. This substantially diversifies the dataset's task scene distribution, ensuring broad coverage of potential scenarios.

# 3 The HumanoidGen Benchmark

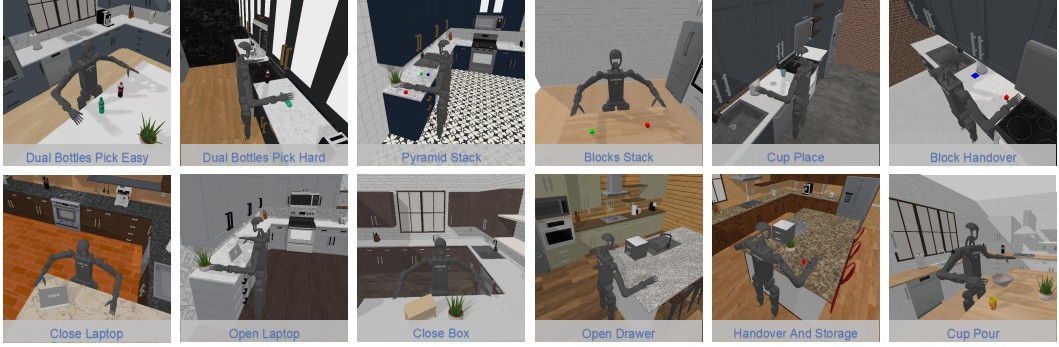

Figure 4: HGen-Bench includes various dexterous bimanual manipulation tasks of varying difficulty. We provide different observation information and deploy the tasks in a home scene.

Based on our framework, we construct a benchmark, **HGen-Bench**, for humanoid manipulation. We designed 20 tasks performed using Inspire hands mounted on the arms of the Unitree H1-2 humanoid robot, which has 7 DoFs for each arm and 6 DoFs for each hand, resulting in a 26-dimensional action space. The examples are given in Fig. 4. In front of the robot, we place a table and construct scenes with various objects, including small items such as blocks of various shapes and colors, articulated objects such as laptops and drawers, and everyday items such as bottles, cups, and plates.

We design atomic operations to fully exploit the dexterity of the hands and define task difficulties from easy to hard settings. We also incorporate bimanual coordination tasks to leverage the capabilities of dual dexterous hands, as well as long-horizon tasks that require geometric reasoning from the LLM planner. To enhance variability, we randomize the initial position, pose, and joint angles of articulated objects within a certain range. We provide RGB and depth images from six camera views, located on both wrists, a first-person perspective, and three third-person views. The details are given in Appendix B.

# 4 Related Works

**LLM-based Data Generation.** Leveraging foundation models to automatically generate diverse tasks and scenes is promising for scaling robotic data sets with minimal human effort. RoboGen [22]

is a generative robotic agent that generates scene components and configurations with language descriptions, and the skills are learned by optimizing LLM-generated reward functions. Gensim [23] adopts LLMs to generate codes to build scenes, simulations, and expert demonstrations. Gensim2 [27] further considers long-term and articulated tasks beyond pick-and-place tasks and incorporates reasoning models for planning. However, these approaches predominantly center on single-arm robots, whereas our work addresses bimanual dexterous manipulation tasks, which are of critical importance for humanoid robots. Regarding expert data collection, LLMs have been leveraged to generate functional control programs for grasp-oriented tasks [28, 29] or articulated object manipulation [30, 31]. In contrast, our method enables code-form planning by framing the manipulation task as a sequence of constraints, thereby eliminating the need for human intervention in defining the functions. Although Rekep [32], OmniManip [33], and RoboTwin [19] utilize spatial reasoning based on key points and axes, they have limitations in handling bimanual dexterous tasks.

**Bimanual Manipulation.** Bimanual dexterous manipulation is promising in solving complex tasks through the coordinated operation of two arms. However, this area faces several challenges, including data scarcity [34, 35], enlarged action spaces [9], diverse collaboration modalities [36, 37], and the intricacies of dexterous hand control [15]. Recent advancements have led to the development of simulation benchmarks for dual-arm systems [21, 19] and humanoid robots [20]. Nonetheless, these benchmarks exhibit limitations in task and scene diversity and typically exclude dexterous hands. These problems motivate us to develop automatic mechanisms for constructing bimanual environments and generating demonstrations, with the aim of broadening the skills and scenarios encompassed within bimanual datasets. For bimanual policy learning, previous methods rely on human-object interaction [38, 39], geometric constraints [40], and arm-movement primitives [41, 42]. In contrast, we do not consider these priors and directly train the 2D and 3D diffusion policies [43, 44], enabling a direct assessment of the effectiveness of the data collection paradigm.

**Datasets and Benchmarks.** Collecting real-world demonstrations is promising for acquiring realistic environmental observations and encompassing the target scenarios of robots. Recent representative datasets, including RT-1 [45], RH-20T [12], DROID [46], Bridge data [13], Open X-Embodiment [10], RoboMind [11], and Agibot World [47], have collected numerous manipulation tasks on specific hardware platforms. However, as we focus on humanoid robots, the embodiment differences make most of the data unsuitable for directly training bimanual dexterous policies. Although methods such as physically interpretable action space [48] and latent action space [2, 49] have been proposed, the adaptation of cross-embodiment data remains an open research challenge. Additionally, although several data augmentation techniques have been introduced [50, 51], the collection of large-scale real-world data remains costly. On the simulation front, benchmarks such as ManiSkill [52], SIMPLER [53], RoboTwin [19], RoboCasa [24], and Garmentlab [54] provide extensive assets and task configurations, while we focus on auto-create task variations and data collection for humanoid robots.

# 5 Experiments

We conducted the following three groups of experiments: (1) a comprehensive evaluation of demonstration generation and execution performance of our framework compared to Robotwin; (2) an effectiveness evaluation of MCTS in enhancing the demonstration generation process; and (3) a validity evaluation of the collected demonstration data in training bimanual dexterous manipulation policies.

In addition to these main experiments, further analyses and evaluations are provided in the Appendix C, including real-world experiments (Appendix C.1), automatic asset annotation evaluation (Appendices C.2), additional challenging dexterous manipulation tasks (Appendix C.3), resource and efficiency analysis of HumanoidGen (Appendix C.4), comparison with existing generation frameworks (Appendix C.5), and comparison across different large models (Appendix C.6).

## 5.1 Evaluation of Data Generation and Execution

**Experimental Setup.** To better show the superiority of our framework, we designed 20 different tabletop manipulation tasks and conducted a quantitative evaluation of the demonstration generation and execution capability of our framework. We divide the 20 tasks into 4 groups based on the following factors: the number of arms used, the length of task horizons, and the complexity of collision scenarios. This categorization allows for a more detailed evaluation of the frameworks'

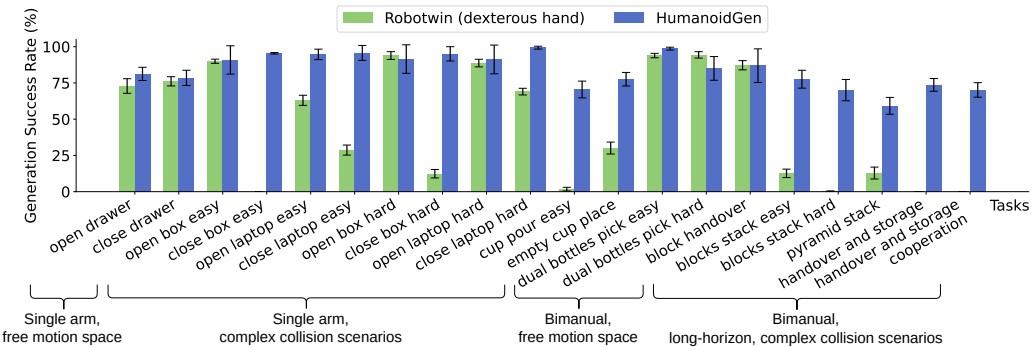

Figure 5: Experiment results of demonstration generation and execution capability. The tasks are categorized into 4 groups, with names based on the number of robotic arms required and their relative difficulty levels. The features of task categories are noted in parentheses.

performance. We compare our method to RobotTwin, where we modify the official implementation to add dexterous hands and provide additional annotations to enable dexterous atomic operations. Besides, we prompt the LLM not to perform proactive dynamical collision management during inference, which reflects the default setup in Robotwin.Using the DeepSeek-R1 [55], we generate and execute the demonstration scripts with both frameworks and compare the final success rates.

**Experimental Results.** As shown in Fig. 5, HumanoidGen demonstrates superior performance across all dexterous manipulation tasks, achieving an average success rate of over 50%. Except for some bimanual long-horizon tasks that involve higher complexity, most task types achieve a success rate above 75%. These results highlight the effectiveness of our framework, making automatic and efficient demonstration generation feasible in bimanual dexterous tasks. In comparison, Robotwin shows comparable performance on single-arm and bimanual short-horizon tasks, indicating that the spatial annotations related to dexterous atomic operations introduced in §2.1 are effective in allowing the dexterous hand to participate in automated demonstration execution for simpler scenarios.

Notably, we observe that HumanoidGen outperforms Robotwin in long-horizon and complex collision tasks, such as the *close & open box* in single-arm tasks and the *blocks stack easy & hard* in bimanual tasks. Further analysis shows that HumanoidGen dynamically manages collision avoidance during inference, handling the collision scenarios flexibly. As an example, in the task *handover and storage*, the collisions with the drawer are appropriately ignored to facilitate trajectory planning for an in-contact pulling operation, while such collisions are taken into account when planning a path around the drawer to retrieve the cube with the right hand. Another example is the *close box* task, where the robot hand circumvents the box lid with collision-awareness to reach an intermediate pose. Subsequently, these collisions are permitted during the flipping operation to complete the closure. These examples demonstrate the potential of our dynamic collision management approach in enabling efficient data collection for long-horizon tasks involving complex collision scenarios.

## 5.2 Effectiveness Evaluation of MCTS

**Experimental Setup.** To evaluate the effectiveness of MCTS in enhancing the demonstration generation process of HumanoidGen, we selected three tasks from §5.1: *blocks stack easy*, *blocks stack hard*, *pyramid stack*, and added a single-arm task, *block stack single*. In contrast to §5.1, we increased the task complexity to rigorously evaluate the performance of LLM in scenarios with insufficiently annotated objects and long-horizon tasks. Specifically, we introduced two key challenges: (i) removing all operation annotations for cubes, forcing the LLM to infer relevant constraints solely from cube poses; (ii) simplifying task descriptions to provide minimal guidance. For example, in the *blocks stack hard* task, the LLM is instructed only to stack cubes into a pile without considering the order of operations and target positions, allowing highly uncertain execution. Building upon this setup, we compared the reasoning success rates and token consumption per execution between MCTS and non-MCTS strategies using DeepSeek-R1 [55] across these four tasks. Additionally, we analyzed the diversity of generated plans to assess the exploratory capabilities of each approach.

| Method | Success rate (%) | Token consumption (K) |
|---|---|---|
| **Block Stack Single** | | |
| Non-MCTS | $63.3 \pm 6.24$ | $15.3 \pm 1.90$ |
| $\text{MCTS}_{N=2}$ | $98.3 \pm 2.36$ | $19.3 \pm 7.04$ |
| **Blocks Stack Easy** | | |
| Non-MCTS | $46.7 \pm 2.36$ | $14.8 \pm 1.64$ |
| $\text{MCTS}_{N=2}$ | $83.3 \pm 5.56$ | $21.6 \pm 7.78$ |
| $\text{MCTS}_{N=3}$ | $95.0 \pm 4.08$ | $22.8 \pm 9.13$ |
| **Blocks Stack Hard** | | |
| Non-MCTS | $18.3 \pm 6.24$ | $16.0 \pm 0.92$ |
| $\text{MCTS}_{N=8}$ | $78.3 \pm 2.36$ | $69.9 \pm 39.24$ |
| $\text{MCTS}_{N=12}$ | $98.3 \pm 2.36$ | $78.3 \pm 51.75$ |
| **Pyramid Stack** | | |
| Non-MCTS | $13.3 \pm 6.24$ | $16.2 \pm 1.29$ |
| $\text{MCTS}_{N=8}$ | $76.7 \pm 4.71$ | $80.0 \pm 31.72$ |
| $\text{MCTS}_{N=12}$ | $90.0 \pm 4.08$ | $89.6 \pm 44.61$ |

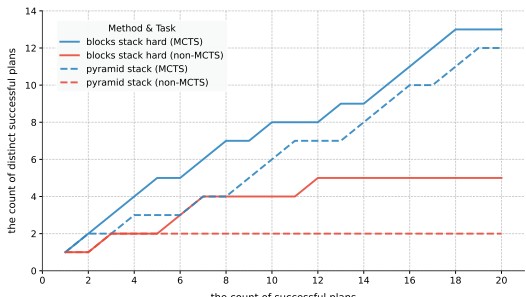

Table 1: The evaluation results of applying different numbers of max MCTS exploration steps $N$ and non-MCTS in four tasks.

Figure 6: The variation of the count of distinct successful plans for MCTS and non-MCTS with the count of successful plans.

**Experimental Results.** As shown in Tab. 1, MCTS can significantly improve the reasoning ability of LLM with minimal additional token consumption. This demonstrates that during the reasoning process of MCTS, LLM can effectively utilize the information of tree nodes to infer the correct task steps. For example, in the 'pyramid stack' task, LLMs are prone to reasoning errors such as unreasonable selections of stacking positions and collisions of two arms in the stacking area. When exploring nodes where such errors occur, MCTS enables LLMs to analyze the causes of errors based on past experiences, correct the execution methods, and continue execution. Further, the results demonstrate that the exploration strategy of MCTS is highly cost-effective, as evidenced across all tasks. For instance, in the *block stack single* task, a 26% increase in token consumption led to a 55% improvement in the reasoning success rate. Specifically, the average token consumption per successful execution method decreased from 24.17K to 19.63K, reducing approximately 20% token consumption. In addition, MCTS can enhance the diversity of generated plans. Fig. 6 illustrates the cumulative number of different plans generated by MCTS and non-MCTS methods as a function of successful planning counts. Among the 20 success solutions, MCTS obtains $\geq 12$ distinct plans, whereas non-MCTS only generated $\leq 5$ distinct plans. This validates that MCTS can explore diverse outcomes by correcting erroneous execution strategies inherent in the direct reasoning approach.

### 5.3 Validity Evaluation of Collected Data

To validate the effectiveness of the collected dataset, we trained policies on the data and evaluated their performance, focusing on how success rates vary with different dataset sizes. We collected 100 data samples for each task using our method. Each sample includes RGB and depth images captured from six cameras, the joint states, and the action ground truth of the robot.

**Policies used for evaluation.** The *Diffusion Policy* (DP) [43] models vision-based robotic control as a conditional denoising process, handling multimodal action distributions and high-dimensional action spaces effectively. The *3D Diffusion Policy* (DP3) [44] enhances DP by incorporating point cloud data, improving performance in dexterous manipulation tasks, and demonstrating strong generalization. We evaluated DP3 and DP on 14 tasks using RGB images or point clouds as input, measuring success rates across different episodes.

**Experimental Results.** As shown in Tab. 2, for some relatively easy tasks, DP3 demonstrates few-shot learning capability. For example, in the *cup pour easy* task, DP3 achieves a 67% success rate using only 20 demonstrations generated by our framework. This validates the diversity of our scene generation and the high quality of our collected data. Overall, DP performs worse than DP3; however, its performance improves with increased data. In the *open box hard* task, the success rate rises from 11.1% with 20 demonstrations to 100% with 100 demonstrations, even surpassing DP3. This confirms that our data scaling approach leads to continuous policy improvement. Notably, some task policies exhibit non-stationary characteristics. Despite achieving a high success rate with 20 demonstrations, their actions exhibit instability and risk, as denoted by the '*' in the table.

For challenging tasks, such as the long-horizon task *blocks stack easy*, both DP and DP3 exhibit a noticeable performance decline, particularly when the data is limited. The suboptimal performance of DP may be attributed to the inherent limitations of RGB information, whereas DP3, which

| Num of Demonstrations | 20 | 50 | 100 | | 20 | 50 | 100 |
|---|---|---|---|---|---|---|---|
| **Blocks Stack Easy** | | | | **Close Drawer** | | | |
| DP3 | $0.0_{\pm 0.0}$ | $0.0_{\pm 0.0}$ | $22.8_{\pm 16.5}$ | DP3 | $83.3_{\pm 17.6}$ | $94.4_{\pm 7.9}$ | $92.6_{\pm 8.3}$ |
| DP | $0.0_{\pm 0.0}$ | $0.0_{\pm 0.0}$ | $0.0_{\pm 0.0}$ | DP | $95.6_{\pm 3.7}$ | $100.0_{\pm 0.0}$ | $100.0_{\pm 0.0}$ |
| **Cup Pour Easy** | | | | **Dual Bottles Pick Easy** | | | |
| DP3 | $67.8_{\pm 10.8}$ | $75.6_{\pm 9.6}$ | $72.2_{\pm 7.9}$ | DP3 | $75.9_{\pm 17.8}$ | $96.3_{\pm 6.9}$ | $93.9_{\pm 7.6}$ |
| DP | $0.0_{\pm 0.0}$ | $2.2_{\pm 6.3}$ | $0.0_{\pm 0.0}$ | DP | $0.0_{\pm 0.0}$ | $0.0_{\pm 0.0}$ | $0.0_{\pm 0.0}$ |
| **Dual Bottles Pick Hard** | | | | **Empty Cup Place** | | | |
| DP3 | $88.9_{\pm 13.6}$ | $90.7_{\pm 11.4}$ | $94.4_{\pm 7.9}$ | DP3 | $25.0_{\pm 8.2}$ | $18.3_{\pm 4.7}$ | $33.3_{\pm 7.1}$ |
| DP | $0.0_{\pm 0.0}$ | $0.0_{\pm 0.0}$ | $0.0_{\pm 0.0}$ | DP | $0.0_{\pm 0.0}$ | $0.0_{\pm 0.0}$ | $6.7_{\pm 13.3}$ |
| **Open Box Easy** | | | | **Open Box Hard** | | | |
| DP3 | $85.6_{\pm 8.0}$ | $95.6_{\pm 4.4}$ | $95.0_{\pm 4.1}$ | DP3 | $95.6_{\pm 5.5}$ | $96.1_{\pm 4.6}$ | $98.3_{\pm 3.3}$ |
| DP | $93.3_{\pm 13.3}$ | $100.0_{\pm 0.0}$ | $100.0_{\pm 0.0}$ | DP | $11.1_{\pm 19.1}$ | $93.3_{\pm 9.4}$ | $100.0_{\pm 0.0}$ |
| **Open Drawer** | | | | **Open Laptop Hard** | | | |
| DP3 | $58.3_{\pm 8.3}$ | $76.0_{\pm 13.1}$ | $84.4_{\pm 11.3}$ | DP3 | $100.0_{\pm 0.0}$ | $100.0_{\pm 0.0}$ | $100.0_{\pm 0.0}$ |
| DP | $17.8_{\pm 22.0}$ | $13.3_{\pm 18.9}$ | $48.9_{\pm 31.4}$ | DP | $15.6_{\pm 22.7}$ | $11.1_{\pm 9.9}$ | $35.6_{\pm 32.4}$ |
| **Close Box Hard** | | | | **Close Laptop Easy** | | | |
| DP3 | $88.9_{\pm 17.6}$ | $96.3_{\pm 6.9}$ | $96.3_{\pm 6.9}$ | DP3 | $100.0_{\pm 0.0}$ | $100.0_{\pm 0.0}$ | $100.0_{\pm 0.0}$ |
| DP | $*82.2_{\pm 22.0}$ | $*51.1_{\pm 19.1}$ | $31.1_{\pm 28.5}$ | DP | $37.8_{\pm 23.9}$ | $40.0_{\pm 23.1}$ | $48.9_{\pm 25.1}$ |
| **Handover and Storage** | | | | **Blocks Stack Hard** | | | |
| DP3 | $0.0_{\pm 0.0}$ | $0.0_{\pm 0.0}$ | $0.0_{\pm 0.0}$ | DP3 | $0.0_{\pm 0.0}$ | $0.0_{\pm 0.0}$ | $0.0_{\pm 0.0}$ |
| DP | $0.0_{\pm 0.0}$ | $0.0_{\pm 0.0}$ | $0.0_{\pm 0.0}$ | DP | $0.0_{\pm 0.0}$ | $0.0_{\pm 0.0}$ | $0.0_{\pm 0.0}$ |

Table 2: We trained DP and DP3 using 100, 50, and 20 trajectories generated by our method, and evaluated the success rates across 14 tasks with 3 random seeds.

requires modeling actions of high DoFs and managing complex long-horizon tasks, necessitates a larger amount of data. Our efficient data collection method enables rapid scaling of data, making it particularly effective for addressing the challenges faced by both DP and DP3 in handling complex long-horizon tasks with limited data. More evaluation details are given in Appendix B.

## 6 Conclusion

This paper introduces HumanoidGen, a framework utilizing automatically generated scenes and synthetic data to facilitate the learning and execution of bimanual dexterous manipulation tasks. Spatial annotations for key points and axes of both assets and hands are given, providing sufficient information for scene generation and planning. The LLM planner generates task decomposition and relational action constraints with collision avoidance applied, enabling the following trajectory optimization of tasks. We also employ collision avoidance and MCTS-based reasoning to improve planning efficiency in complex or long-horizon tasks. We construct a benchmark that contains diverse bimanual dexterous tasks for evaluation. The experiments show that our framework has superior capabilities in demonstration script generation and execution, especially for tasks with long task horizons and complex collision scenarios. MCTS improves LLMs' reasoning ability with insufficient annotations. The training of diffusion policies verifies the quality and scaling capacity of our method.

## Acknowledgments

This work is supported by the National Natural Science Foundation of China (Grant Nos. 62427819 and 62306242), the Young Elite Scientists Sponsorship Program by CAST (Grant No. 2024QNRC001), the Yangfan Project of the Shanghai (Grant No.23YF11462200), and the Science and Technology Commission of Shanghai Municipality (No. 24511103100).

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

# Appendix Table of HumanoidGen

# A   Implementation Details of HumanoidGen

## A.1   Automatic Asset Annotation

Figure 7: The illustration of automatic asset annotation.

In this section, we detail how to simplify the annotation process using Stable Diffusion to automatically annotate similar assets and avoid repetitive annotation work. As shown in Fig. 7, assets of the same category have similar annotation information (points and axes), enabling direct annotation migrations across them. Specifically, we utilize the Stable Diffusion encoder for feature point matching. First, from the same viewing angle where key

points are not occluded, we obtain the image $I^a$ of the annotated asset and the image $I^{\mathrm{non}}$ of the unannotated asset. For the annotated asset, there are $n$ key points $P^a = \{p_1^a, p_2^a, ..., p_n^a\}$, and our goal is to obtain the corresponding key points $P^{\mathrm{non}}$ in $I^{\mathrm{non}}$. Following the method in [19], we extract the diffusion features of both $I^a$ and $I^{\mathrm{non}}$. For each $p_i^a \in P^a$, which corresponds to a single pixel in $I^a$, we analyze the similarity of the extracted diffusion features to obtain the corresponding pixel in $I^{\mathrm{non}}$. Since the images are captured without occlusion, we can obtain $p_i^{\mathrm{non}} \in P^{\mathrm{non}}$ in sequence by back-projecting the pixels in $I^{\mathrm{non}}$ into the 3D asset space. For axes annotation, starting from the obtained key points, the same axis directions as the original asset are extended to obtain the key axes of the unannotated asset. By applying the approach above, the key points and key axes of the new asset are automatically annotated, effectively reducing the necessary effort by manual annotation.

## A.2 Constraint Solver

As explained in §2.2, the LLM planner decomposes the long-horizon task into a sequence composed of hand atomic operations and arm movements. The arm movement solving problem is formulated as two constrained optimization problems, with the constraints $C^{\mathrm{goal}}$ and $C^{\mathrm{path}}$ both inferred by the LLM. We consider a movement process to consist of $T$ time steps. Here, $t \in \{0, 1, ..., T\}$ represents each time step, with $t = 0$ and $t = T$ denoting the initial and the end time step, respectively. At each $t$, the arm joint angle is $\theta_t$, and the pose of the end effector is $\mathbf{e}_t \in SE(3)$.

The motion solving process involves two steps. The first step is to (i) obtain the target end-effector pose $\mathbf{e}_T$ and the target arm angle $\theta_T$. This can be formulated as a nonlinear optimization problem as

$$
\arg\min_{\theta_T} \sum_{c_i \in C^{\mathrm{goal}}} w_i c_i(u_{\mathrm{act}}, u_{\mathrm{all}}) + w_{\mathrm{reg}} \|\theta_T - \theta_{\mathrm{nominal}}\|,
$$

$$
\text{s.t.} \quad
\begin{cases}
\mathbf{e}_T = f_{\mathrm{FK}}(\theta_T) & \text{(Kinematic constraints)} \\
c(u_{act}, u_{all}) \in [c_{\mathrm{lower}}, c_{\mathrm{upper}}], \forall c \in C^{\mathrm{goal}} & \text{(Relational action constraints)} \\
\theta_T \in \Theta_{\mathrm{collision\_free}} & \text{(Collision avoidance constraints)}
\end{cases}, \quad (1)
$$

where $w$ denotes the weight to balance various optimization terms. $c(u_{\mathrm{act}}, u_{\mathrm{all}})$ represents a relational constraint between $\mathcal{F}_{\mathrm{act}}$ and $\mathcal{F}_{\mathrm{all}}$, defined in §2.2. Here, $u \in \{p, v, l(p, v)\}$ is used as a basic component for formulating the constraints, with $p$, $v$, and $l(p, v)$ denoting points, vectors, and lines, respectively. Thus, $c(u_{\mathrm{act}}, u_{\mathrm{all}})$ can be further expressed as

$$
c(u_{\mathrm{act}}, u_{\mathrm{all}}) =
\begin{cases}
d(\mathbf{e}_T \cdot p_{\mathrm{act}}, p_{\mathrm{all}}) & \text{(Distance between points)} \\
d(\mathbf{e}_T \cdot p_{\mathrm{act}}, l(p_{\mathrm{all}}, v_{\mathrm{all}})) & \text{(Distance between point and line)} \\
d(l(\mathbf{e}_T \cdot p_{\mathrm{act}}, \mathbf{e}_T \cdot v_{\mathrm{act}}), p_{\mathrm{all}}) & \text{(Distance between line and point)} \\
a(v_{\mathrm{act}}, v_{\mathrm{all}}) & \text{(Angle difference between axes)}
\end{cases}, \quad (2)
$$

where $l(p, v)$ is the line formed by the point $p$ and the axis $v$, $d(\cdot)$ is the function to calculate the distance between points or between a point and a line, and $a(\cdot)$ is the function to calculate the angle difference between two axes. $w_{\mathrm{reg}} \|\theta_T - \theta_{\mathrm{nominal}}\|$ represents the regularization term, which biases the optimization toward the robot's safe configuration. The nominal angles $\theta_{\mathrm{nominal}}$ can generally be predefined using historical data or expert knowledge, such as $\theta_0$ and $\theta_{\mathrm{default}}$. The pose $\mathbf{e}_T$ of the end-effector must satisfy the constraints derived from the forward kinematics solution $f_{\mathrm{FK}}(\theta_T)$. The interval $[c_{\mathrm{lower}}, c_{\mathrm{upper}}]$ represents the upper and lower bounds of the relational constraints. Finally, $\Theta_{\mathrm{collision\_free}}$ denotes the set of $\theta$ values that will not result in collisions. We employed the SNOPT solver from the pydrake library to solve this optimization problem.

The second step is to (ii) calculate $\mathbf{e}_t$ and $\theta_t$, when $0 < t < T$. To address this problem, we utilize the constrained motion planner from the mplib library to minimize the cost function $\mathrm{Cost}(\theta_t)$ while guaranteeing the satisfaction of the constraints during the movement process. The optimization problem can be formulated as

$$
\arg\min_{\theta_{t \in [1, T-1]}} \mathrm{Cost}(\theta_t), \quad \text{s.t.}
\begin{cases}
\mathbf{e}_t = f_{\mathrm{FK}}(\theta_t), \forall t \in [1, T-1] & \text{(Kinematic constraint)} \\
c(u_{act}, u_{all}) \in [c_{\mathrm{lower}}, c_{\mathrm{upper}}], \forall c \in C^{\mathrm{path}} & \text{(Relational action constraints)}, \\
\theta_t \in \Theta_{\mathrm{collision\_free}}, \forall t \in [1, T-1] & \text{(Collision avoidance)}
\end{cases}
$$
$$(3)$$

where $\mathrm{Cost}(\theta_t)$ typically incorporates objectives related to motion smoothness, energy efficiency, and trajectory optimality.

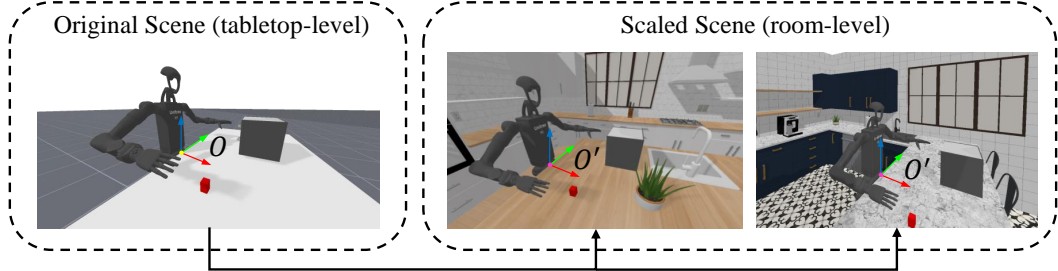

Figure 8: Schematic diagram illustrating the scene scaling of the *handover and storage* task, extending from a desktop-level scenario on the left to a room-level scenario on the right, aimed at enhancing scene diversity in the dataset.

## A.3  Scene Scaling

To scale tasks from tabletop-level scenes to room-level scenes, as introduced in §2.3, coordinate transformations are applied to align the scene configurations. Specifically, in the original scene, a point is selected on the table edge near the robot as the origin $o$ to construct a Cartesian coordinate system $O$–xyz. The x-axis of $O$ points in the direction the robot faces toward the table, the y-axis extends from $o$ along the table's edge toward the robot's left-hand direction, and the z-axis extends vertically upward from $o$. Correspondingly, in the new scene, an edge point in an open desktop area is selected as the origin $o'$ to construct another Cartesian coordinate system $O'$–xyz, with x, y, and z directions consistent with those in $O$–xyz. Thus, for a point $p$, vector $v$, and pose $\mathcal{P} \in SE(3)$ in the world coordinate system of the original scene, and their counterparts $p'$, $v'$, and $\mathcal{P}' \in SE(3)$ in world coordinate system of the scaled scene, the following relationship holds:

$$\alpha' = T_{O \to O'}\, \alpha, \quad \text{where} \quad \alpha \in \{p, v, \mathcal{P}\},\ \alpha' \in \{p', v', \mathcal{P}'\}. \tag{4}$$

Based on the coordinate transformation defined in Eq. (4), the spatial relationships of objects, robots, key points, and axes in the original system $O$ can be directly mapped to the target system $O'$. This allows reconstructed poses (e.g., object configurations, relational action constraints) to inherit geometric consistency from the source scene, enabling seamless task execution in the scaled environment. By leveraging this lightweight approach, our dataset diversity is enhanced without requiring additional annotations or complex geometric reasoning.

## A.4  MCTS-Based LLM Reasoning

**Selection.**  In each iterative loop, the step begins with a selection to find the node to expand. In the selection step, it starts from the root node and explores downwards. When a node is explored, the algorithm decides whether to continue exploring one of its children or to end the exploration and select the node for expansion. To define the action to expand from the explored node, we introduce a special branch $S' = \emptyset$ as proposed in [25]. Here, $S'$ is the branch defined in §2.3, which consists of multiple operation steps $S$ with consistent intents. The choice of continuing exploration and expansion is decided by

$$\text{SelectPolicy}(n) = \underset{S' \in \text{Children}(n) \cup \{\varnothing\}}{\arg\max}\ Q_{\text{DUCB}}(n, S'), \tag{5}$$

where we use $Q_{\text{DUCB}}$ as the value estimation method to balance the values of exploration and exploitation, as referenced in [25]. $Q_{\text{DUCB}}$ uses a discount $\gamma \in (0, 1)$ to smoothly drop those outdated feedback records and give greater weights to the feedback from the latest backpropagation during value estimation. The process can be described as

$$Q_{DUCB}(n, S') = \frac{W_\gamma(n, S')}{N_\gamma(n, S')} + \sqrt{\frac{2 \ln \sum_{S'' \in \text{Children}(n) \cup \{\varnothing\}} N_\gamma(n, S'')}{N_\gamma(n, S')}}, \tag{6}$$

$$W_\gamma(n, S') = \sum_{t=1}^{N(n,S')} \gamma^{N(n,S')-t} R(\tau_t), \tag{7}$$

$$N_\gamma(n, S') = \sum_{t=0}^{N(n,S')-1} \gamma^t, \tag{8}$$

$$N(n, S') = |\Gamma(n, S')|, \tag{9}$$

where $\tau_t = \{(n_{\mathrm{root}}, n^{(1)}), (n^{(1)}, n^{(2)}), ..., (n^{(|\tau|-1)} = n_t, \varnothing)\}$ denotes the selection trajectory of t-th iteration that ends with $n_t$ as the expanding node. $\Gamma(n, S') = \{\tau | (n, S') \in \tau\}$ denotes the set of $\tau$ that contains $(n, S')$. $R(\tau)$ represents the value of the trajectory $\tau$, which is determined in the backpropagation stage. $N(n, S')$ denotes the number of trajectories $\tau$ that contain $(n, S')$. $W(n, S')$ denotes the total value of trajectories $\tau$ that contain $(n, S')$. $N_\gamma(n, S')$ and $W_\gamma(n, S')$ are both results decayed by $\gamma$.

**Expansion.** Using the information stored in the exploration node, such as executed code, non-executable code, and the node's runtime scene information, a prompt is constructed and fed into the reasoning LLM. Then the LLM infers new executable code. Following the STCR mechanism proposed in §2.3, a subtree is built with the selected expansion node as the root. When expanding, if the next execution branch $S'_{\mathrm{new}}$ under the current expansion node $n_{\mathrm{now}}$ already exists, i.e., $S'_{\mathrm{new}} \in \mathrm{Children}(n_{\mathrm{now}})$, a new node will not be generated, and the expansion will continue along the existing branch. If the task is successfully completed, the MCTS process ends, and the executed code is concatenated to the generated code to form the final successfully executed code. If the task remains incomplete, the iterative loop continues.

**Backpropagation.** Backpropagation is the final step of the iteration. It aims to update the reward $R(\tau)$, where $\tau$ is the selected trajectory in this iteration. Since iteration ends once a successful task plan is discovered, the extrinsic reward contains $R_{\mathrm{extrinsic}} = 0$ during the iteration process and cannot be used as a reward. Therefore, we propose an intrinsic exploration value $R_{\mathrm{intrinsic}}$ based on 'valuable moment', which refers to the occurrence of a valuable event, such as 'successfully grasp a cup', 'pinch a cube', 'the object does not fall off when the hand is opened', etc. If the generated code contains at least one 'valuable moment' during execution, then we set $R_{\mathrm{intrinsic}} = 1$ for backpropagation to incentivize the expansion of the LLM.

# B    Details of HGen-Bench

**Benchmark Task Design.** Our benchmark builds on ManiSkill3 [52] physics engine. In the simulation, the maximum control frequency can be set to 100 Hz, and the maximum camera capture frequency is 100 Hz. Our tasks span various scenarios. The details are given as follows. (i) *Atom operations*. To fully leverage the dexterity of the hands, we design interaction modes such as pinching and grasping. For example, stacking small blocks requires pinching them with the index finger and thumb, whereas picking up a bottle requires a firm grasp. (ii) *Task difficulties*. To reflect varying levels of task difficulty, we define difficulty settings and name the tasks with 'easy' and 'hard' to distinguish. For instance, in the *dual bottles pick* task, the 'easy' setting involves picking upright bottles, while the 'hard' setting involves picking fallen ones. (iii) *Collaboration modes*. We also design bimanual coordination tasks that exploit the capabilities of dual dexterous hands. For example, the *block handover* task involves picking up a block, handing it over from hand to hand, and placing it. (iv) *Long-horizon and geometric reasoning*. We design a set of challenging tasks to evaluate these capabilities. For example, the long-horizon task *handover and storage* involves a sequence of five steps: the right hand grasps a block and positions it for transfer, the left hand opens a drawer, then takes over the block, and finally places it into the drawer. We also design tasks requiring geometric reasoning. In *blocks stack hard*, the robot must stack three blocks vertically. In *pyramid stack*, two blocks must be placed on the first layer, with a third block stacked on top to form the second layer. To introduce variability, we randomize the initial position, pose, and joint angles of articulated objects within a certain range. This allows us to collect more diverse data and test the generalization ability of learned policies.

**Policy Training Examples.** We provide example code for deploying and evaluating different types of policies. In our experiments, we present the training and evaluation results of these policies. The dataset we collected includes first-person RGB images captured by an Intel RealSense D435 camera, aligned depth maps, and point clouds generated using Open3D, as well as joint angles as part of the observation data. We use first-person RGB images and joint angles as inputs to DP. For DP3, we crop the point cloud to retain only the workspace, i.e., the hand, arm, and the object to be manipulated. The point cloud is then downsampled to 1024 points. This sparse point cloud, together with the joint angles, is used as the input to DP3.

We train both DP and DP3 using 100, 50, and 20 trajectories generated by our method. During each training session, we record results from three checkpoints. For DP, we evaluate checkpoints at epochs 150, 200, and 250, while for DP3, we evaluate at epochs 1500, 2000, and 2500. For evaluation, we test each of the three checkpoints using seeds 0, 1, and 2, and report the mean and standard deviation of the success rates.

The full success rate results across 20 tasks are shown in Tab. 3. For relatively simple tasks, DP3 exhibits few-shot learning capabilities when trained with data collected using our method, achieving high success rates with as few as 20 trajectories. For moderately difficult tasks where 20 trajectories yield insufficient performance, increasing the number of trajectories consistently improves success rates, indicating the effectiveness of our data scaling strategy. However, for several extremely challenging tasks, although our data generation method demonstrates high-quality demonstrations, neither DP nor DP3 is able to accurately learn each step. These tasks often involve long-horizon sequences and fine-grained object manipulation. We anticipate that future methods

Table 3: We present the DP and DP3 results for all 20 tasks using 100, 50, and 20 trajectories generated by our method, and evaluate the success rates across 14 tasks using 3 random seeds.

| Num of Demonstrations | 20 | 50 | 100 | | 20 | 50 | 100 |
|---|---|---|---|---|---|---|---|
| **Blocks Stack Easy** | | | | **Close Drawer** | | | |
| DP3 | $0.0_{\pm0.0}$ | $0.0_{\pm0.0}$ | $22.8_{\pm16.5}$ | DP3 | $83.3_{\pm17.6}$ | $94.4_{\pm7.9}$ | $92.6_{\pm8.3}$ |
| DP | $0.0_{\pm0.0}$ | $0.0_{\pm0.0}$ | $0.0_{\pm0.0}$ | DP | $95.6_{\pm3.7}$ | $100.0_{\pm0.0}$ | $100.0_{\pm0.0}$ |
| **Cup Pour Easy** | | | | **Dual Bottles Pick Easy** | | | |
| DP3 | $67.8_{\pm10.8}$ | $75.6_{\pm9.6}$ | $72.2_{\pm7.9}$ | DP3 | $75.9_{\pm17.8}$ | $96.3_{\pm6.9}$ | $93.9_{\pm7.6}$ |
| DP | $0.0_{\pm0.0}$ | $2.2_{\pm6.3}$ | $0.0_{\pm0.0}$ | DP | $0.0_{\pm0.0}$ | $0.0_{\pm0.0}$ | $0.0_{\pm0.0}$ |
| **Dual Bottles Pick Hard** | | | | **Empty Cup Place** | | | |
| DP3 | $88.9_{\pm13.6}$ | $90.7_{\pm11.4}$ | $94.4_{\pm7.9}$ | DP3 | $25.0_{\pm8.2}$ | $18.3_{\pm4.7}$ | $33.3_{\pm7.1}$ |
| DP | $0.0_{\pm0.0}$ | $0.0_{\pm0.0}$ | $0.0_{\pm0.0}$ | DP | $0.0_{\pm0.0}$ | $0.0_{\pm0.0}$ | $6.7_{\pm13.3}$ |
| **Open Box Easy** | | | | **Open Box Hard** | | | |
| DP3 | $85.6_{\pm8.0}$ | $95.6_{\pm4.4}$ | $95.0_{\pm4.1}$ | DP3 | $95.6_{\pm5.5}$ | $96.1_{\pm4.6}$ | $98.3_{\pm3.3}$ |
| DP | $93.3_{\pm13.3}$ | $100.0_{\pm0.0}$ | $100.0_{\pm0.0}$ | DP | $11.1_{\pm19.1}$ | $93.3_{\pm9.4}$ | $100.0_{\pm0.0}$ |
| **Open Drawer** | | | | **Open Laptop Hard** | | | |
| DP3 | $58.3_{\pm8.3}$ | $76.0_{\pm13.1}$ | $84.4_{\pm11.3}$ | DP3 | $100.0_{\pm0.0}$ | $100.0_{\pm0.0}$ | $100.0_{\pm0.0}$ |
| DP | $17.8_{\pm22.0}$ | $13.3_{\pm18.9}$ | $48.9_{\pm31.4}$ | DP | $15.6_{\pm22.7}$ | $11.1_{\pm9.9}$ | $35.6_{\pm32.4}$ |
| **Close Box Hard** | | | | **Close Laptop Easy** | | | |
| DP3 | $88.9_{\pm17.6}$ | $96.3_{\pm6.9}$ | $96.3_{\pm6.9}$ | DP3 | $100.0_{\pm0.0}$ | $100.0_{\pm0.0}$ | $100.0_{\pm0.0}$ |
| DP | $*82.2_{\pm22.0}$ | $*51.1_{\pm19.1}$ | $31.1_{\pm28.5}$ | DP | $37.8_{\pm23.9}$ | $40.0_{\pm23.1}$ | $48.9_{\pm25.1}$ |
| **Handover and Storage** | | | | **Blocks Stack Hard** | | | |
| DP3 | $0.0_{\pm0.0}$ | $0.0_{\pm0.0}$ | $0.0_{\pm0.0}$ | DP3 | $0.0_{\pm0.0}$ | $0.0_{\pm0.0}$ | $0.0_{\pm0.0}$ |
| DP | $0.0_{\pm0.0}$ | $0.0_{\pm0.0}$ | $0.0_{\pm0.0}$ | DP | $0.0_{\pm0.0}$ | $0.0_{\pm0.0}$ | $0.0_{\pm0.0}$ |
| **Block Handover** | | | | **Close Box Easy** | | | |
| DP3 | $0.0_{\pm0.0}$ | $0.0_{\pm0.0}$ | $0.0_{\pm0.0}$ | DP3 | $100.0_{\pm0.0}$ | $98.3_{\pm3.3}$ | $99.4_{\pm1.6}$ |
| DP | $0.0_{\pm0.0}$ | $0.0_{\pm0.0}$ | $0.0_{\pm0.0}$ | DP | $97.8_{\pm6.3}$ | $100.0_{\pm0.0}$ | $91.1_{\pm13.7}$ |
| **Close Laptop Hard** | | | | **Handover and Storage Cooperation** | | | |
| DP3 | $92.6_{\pm8.3}$ | $94.4_{\pm7.9}$ | $96.3_{\pm6.9}$ | DP3 | $0.0_{\pm0.0}$ | $0.0_{\pm0.0}$ | $0.0_{\pm0.0}$ |
| DP | $*46.7_{\pm13.3}$ | $*42.2_{\pm34.6}$ | $33.3_{\pm26.7}$ | DP | $0.0_{\pm0.0}$ | $0.0_{\pm0.0}$ | $0.0_{\pm0.0}$ |
| **Open Laptop Easy** | | | | **Pyramid Stack** | | | |
| DP3 | $71.1_{\pm5.7}$ | $77.2_{\pm7.5}$ | $81.1_{\pm9.4}$ | DP3 | $0.0_{\pm0.0}$ | $0.0_{\pm0.0}$ | $0.0_{\pm0.0}$ |
| DP | $*75.6_{\pm18.3}$ | $*71.1_{\pm16.6}$ | $60.0_{\pm16.3}$ | DP | $0.0_{\pm0.0}$ | $0.0_{\pm0.0}$ | $0.0_{\pm0.0}$ |

that are capable of learning long-horizon behaviors will help address these challenges. As discussed in the main text, during DP training, some tasks show abnormally high success rates even when the policy has not learned a valid strategy. We mark such results with '*' in the table. For example, in the task *Close Box Hard*, with only 20 demonstrations, the arm tends to make rapid and random movements, occasionally closing the box lid by chance, which results in a deceptively high success rate. However, this behavior is unstable and unsafe. As the number of demonstrations increases (e.g., at 100 demonstrations), the arm no longer moves randomly and instead attempts to perform a deliberate closing motion. In this case, the hand needs to approach the lid from behind. Due to DP's lack of spatial understanding and perception, the hand often collides with the lid, failing to maneuver behind it and ultimately leading to lower success rates.

**Decision.** We adopt a diffusion policy framework for both DP and DP3, which predicts a future sequence of $H$ actions conditioned on $n_{\mathrm{obs}}$ steps of past observations. At inference time, the last $N_{\mathrm{act}} = H - n_{\mathrm{obs}} + 1$ predicted actions are executed to form the final control trajectory. In our experiments, we set $H = 8$ and $n_{\mathrm{obs}} = 3$ for both DP3 and DP. While both DP and DP3 share the core principle of conditional denoising diffusion, they differ in input modalities and policy architectures.

DP3 processes 3D point cloud inputs using a DP3 encoder, where each frame consists of 1024 points with 3D coordinates. The encoded feature has a dimension of 128. The architecture incorporates Feature-wise Linear Modulation (FiLM) at the down, mid, and up layers of the convolutional UNet to improve conditional representation learning. In contrast, DP operates on image-based observations alongside low-dimensional proprioceptive inputs. The visual encoder is implemented using the `MultiImageObsEncoder` module, which supports multiple RGB observation keys, each associated with its own ResNet18 backbone (without pre-trained weights) or a shared model depending on configuration.

Both methods adopt a noise schedule with $\beta_{\mathrm{start}} = 0.0001$, $\beta_{\mathrm{end}} = 0.02$, and a squared cosine schedule (`squaredcos_cap_v2`) over 100 diffusion training steps. However, the sampling strategies differ: DP3 employs DDIM (`prediction_type=sample`), while DP uses DDPM with $\epsilon$-prediction (`prediction_type=epsilon`) and enables `clip_sample` to stabilize training.

**Training Setting.** For both DP and DP3, we adopt an exponential moving average (EMA) to stabilize the training process. The EMA parameters are updated with `inv_gamma = 1.0`, `power = 0.75`, and `max_value = 0.9999`. Since BatchNorm is not compatible with EMA, all normalization layers are replaced with GroupNorm to ensure training stability, particularly for DP with image inputs. Both policies use the AdamW optimizer with a learning rate of $1 \times 10^{-4}$, $\beta = [0.95, 0.999]$, and a weight decay of $1 \times 10^{-6}$. A cosine learning rate scheduler with a linear warm-up of 500 steps is applied to improve convergence during the initial training phase.

Table 4: Randomization Settings

| | Position Randomization | Angle Randomization | | Position Randomization | Angle Randomization |
|---|---|---|---|---|---|
| Blocks Stack Easy | [-0.005, 0.005, -0.02, 0.02] | [-15, 15] | Close Drawer | [-0.05, 0.05, 0.18, -0.18] | [-25, 25] |
| Cup Pour Easy | [-0.02, 0.02, -0.04, 0.04] | [-30, 30] | Dual Bottles Pick Easy | [-0.005, 0.035, -0.025, 0.035] | [0, 0] |
| Dual Bottles Pick Hard | [-0.005, 0.035, -0.025, 0.035] | [0, 0] | Empty Cup Place | [-0.02, 0.02, -0.04, 0.04] | [-30, 30] |
| Open Box Easy | [-0.02, 0.02, -0.04, 0.04] | [30, 30] | Open Box Hard | [-0.02, 0.02, -0.04, 0.04] | [30, 30] |
| Open Drawer | [-0.05, 0.05, -0.18, 0.18] | [-25, 25] | Open Laptop Hard | [-0.02, 0.02, -0.04, 0.04] | [30, 30] |
| Close Box Hard | [-0.05, 0.05, -0.18, 0.18] | [-25, 25] | Close Laptop Easy | [-0.05, 0.05, -0.18, 0.18] | [-25, 25] |
| Handover and Storage | [-0.01, 0.01, 0, 0] | [0, 0] | Blocks Stack Hard | [-0.005, 0.005, -0.02, 0.02] | [-15, 15] |
| Block Handover | [-0.005, 0.005, -0.005, 0.005] | [0, 0] | Close Box Easy | [-0.05, 0.05, -0.18, 0.18] | [-25, 25] |
| Close Laptop Hard | [-0.05, 0.05, -0.18, 0.18] | [-25, 25] | Handover and Storage Cooperation | [-0.01, 0.01, 0, 0] | [0, 0] |
| Open Laptop Easy | [-0.01, 0.01, 0.04, -0.04] | [-30, 10] | Pyramid Stack | [-0.02, 0.02, -0.04, 0.04] | [-30, 30] |

DP3 is trained for 3000 epochs with checkpoints saved every 500 epochs. In contrast, DP is trained for 300 epochs with checkpoints saved every 50 epochs. The batch size is set to 256 for DP3 and 64 for DP, reflecting the difference in memory requirements between point cloud and image inputs. Data loading includes shuffling for training and deterministic ordering for validation, with `pin_memory=True` to improve performance.

**Randomization Setting.** In the data collection and policy performance evaluation, We defined the randomization ranges for object positions and orientations, as shown in Tab. 4. Specifically, *Position Randomization* $[x_1, x_2, y_1, y_2]$ (in meters) denotes the range of randomized positions, where $x_1$ and $x_2$ represent the bounds for the $x$-coordinate, and $y_1$ and $y_2$ represent the bounds for the $y$-coordinate. *Angle Randomization* $[\theta_1, \theta_2]$ (in degrees) indicates the range for orientation randomization. To accommodate the capabilities of different algorithms, a small subset of tasks (DP3 vs. DP) adopts slightly different randomization ranges.

## C Experimental Details and Additional Experiments

### C.1 Real-World Experiments

To further explore the application of HumanoidGen in real-world settings, we conducted experiments focusing on both data generation and sim2real transfer.

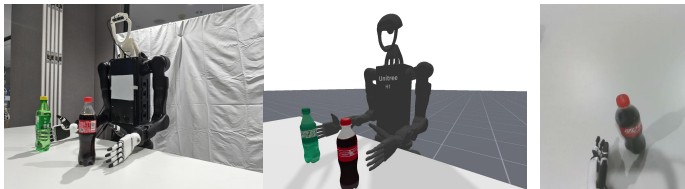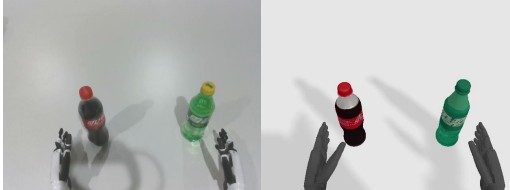

Figure 9: Visualization of real scene and simulation scene.

**Experiment Setup.** As shown in Fig. 9, we set up a real-world environment that closely mirrors the simulation environment. This includes a white table and a humanoid robot performing bimanual dexterous manipulation tasks. The robot used is the Unitree H1-2, equipped with a D435i depth camera mounted on its head, with a resolution of 640×480. The robot's dexterous hands are the same Inspire hands used in the simulation environment. For the tasks, we selected the *dual bottles pick easy* and *close laptop hard* from the HGen-Bench for real-world transfer, using physical assets that match their simulated counterparts. The object poses in each trial were randomized to test robustness. Specifically, for the *dual bottles pick easy*, the placement was randomized within a range of 6 cm × 6 cm and ±15° in orientation, while for the *close laptop hard*, the randomization range was 6 cm × 6 cm and +10° to +20° in orientation.

**Experimental Procedure.** The experiment consists of three main steps: **(i) Real2Sim Pose Estimation.** Using the existing digital assets, we first estimate the current poses of objects in the camera coordinate system with FoundationPose [56]. The estimated 6D poses are then transformed into the world coordinate system, allowing the corresponding digital assets to be placed at the same poses in the simulation environment. **(ii) Trajectory Collection.** We leverage LLM reasoning and the trajectory generation module of our framework to automatically generate execution trajectories in simulation, which are then executed on the real robot to collect real-world data. For each task, we collected 25 real-world trajectories. **(iii) Policy Training and Sim2Real Transfer.** To evaluate the effectiveness of the collected real-world data, we train diffusion policy models using 5 and 25 real-world trajectories, respectively. In addition, to investigate the contribution of simulation data to real-world policy performance, we augment the real-world dataset with additional simulated trajectories. Specifically, for each task, we train and evaluate policies using 5 real-world trajectories combined with 100 simulated trajectories, aiming to analyze how simulation data improves real-world performance.

Table 5: **Experiment Results.** DP represents the diffusion policy success rate under different training settings: 5 real trajectories (5R), 25 real trajectories (25R), and 100 simulated + 5 real trajectories (100S+5R).

| Task | Collection Success Rate (%) | Collection Time (s) | DP (5R) | DP (25R) | DP (100S+5R) |
|------|------|------|------|------|------|
| **Dual Bottles Pick Easy** | 85 | 16 | 0% (0/20) | **60% (12/20)** | **65% (13/20)** |
| **Close Laptop Hard** | 90 | 13 | 20% (4/20) | **70% (14/20)** | **70% (14/20)** |

**Experimental Results.** We evaluated two tasks, *dual bottles pick easy* and *close laptop hard*, in real-world experiments. The metrics include the data collection success rate, the average collection time per trajectory, and the performance of DP [43] models trained under different data settings. As shown in Tab. 5, the average time to collect a single real-world trajectory is only about 14 seconds, and the data collection success rate exceeds 85% for both tasks. The results of DP model training demonstrate that policies trained with 25 real-world trajectories significantly outperform those trained with only 5. This indicates that the success rate increases with the amount of real-world data. Furthermore, pretraining with 100 simulated trajectories followed by fine-tuning on 5 real-world trajectories achieves comparable or even better performance than using 25 real-world trajectories alone, particularly in the *dual bottles pick easy* task.

## C.2 Automatic Asset Annotation Evaluation

To assess the effectiveness of our automatic annotation method based on annotation migration, we conducted quantitative analyses.

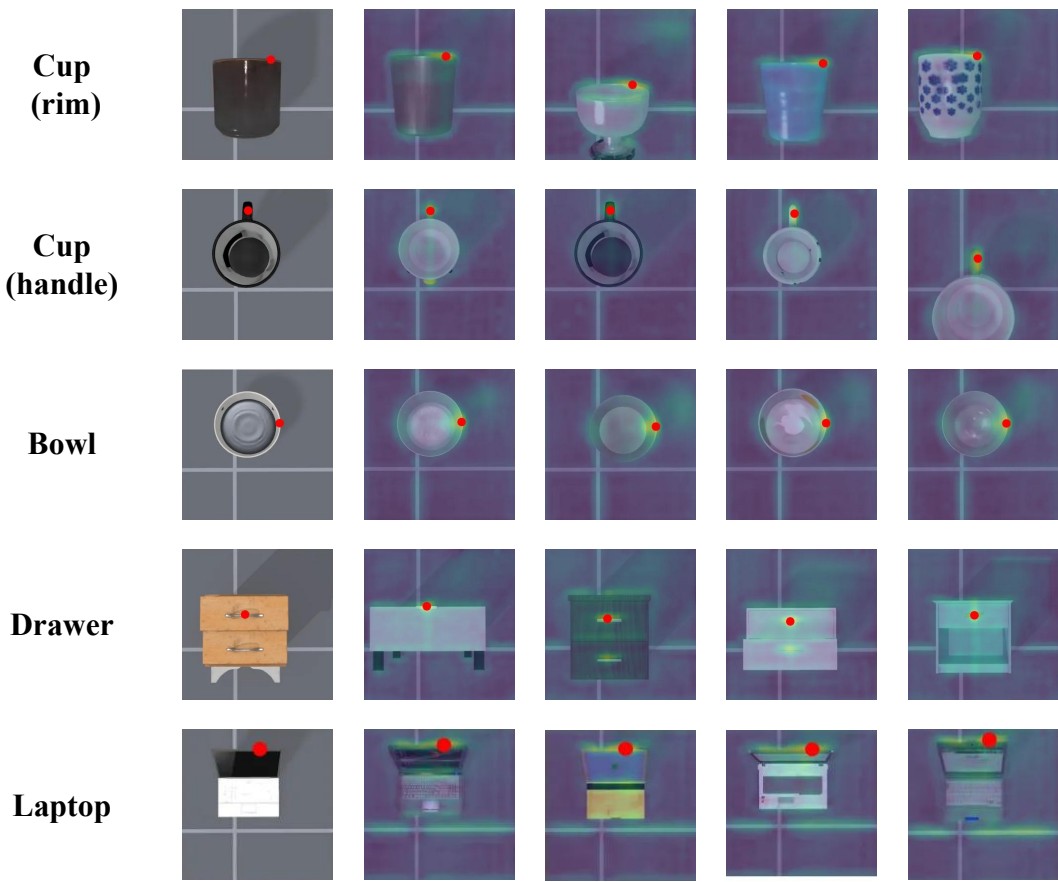

Figure 10: Visualization of annotation migration.

**Experimental Setup.** As illustrated in Fig. 10, we selected five parts from four commonly used object types for automatic annotation. The cup contains two parts: the rim and the handle. For each object category, 20 assets were randomly selected from [14, 24, 19]. The rendering and reprojection processes for automatic annotation were implemented using ManiSkill3 [52], the same simulation environment as our data generation framework.

Table 6: Evaluation of annotation time and success rate.

| Evaluation Metric | Cup (rim) | Cup (handle) | Bowl | Drawer | Laptop |
|---|---|---|---|---|---|
| **Manual Annotation Time (s)** | 4.00 | 4.00 | 4.00 | 6.00 | 6.00 |
| **Automated Migration Time (s)** | 0.72 | 0.72 | 0.79 | 0.80 | 0.81 |
| **Migration Success Rate (%)** | 100.0% (20/20) | 90.0% (18/20) | 100.0% (20/20) | 90.0% (18/20) | 100.0% (20/20) |

**Experimental Results.** We compared the annotation time between manual and automatic annotation and evaluated the success rate of automatic annotation, as summarized in Tab. 6. For manual annotation, simple rigid objects such as cups and bowls required about 4 seconds per part, while articulated objects like laptops and drawers took around 6 seconds. In contrast, our automatic annotation required only about 0.7 seconds per asset on average, achieving a substantial acceleration. The success rate of automatic annotation was also high, reaching 100% for the cup rim, bowl, and laptop, and over 90% for other parts. The few failures, such as those for the cup handle and drawer, were primarily caused by significant geometric variations, such as differences in handle shapes between circular and rectangular designs.

## C.3 Additional Challenging Dexterous Manipulation Tasks

To further evaluate the scalability and robustness of our framework beyond the 20 main tabletop manipulation tasks, we design five additional challenging tasks focusing on bimanual coordination, dexterous contact-rich operations, and complex tabletop environments, as illustrated in Fig. 11.

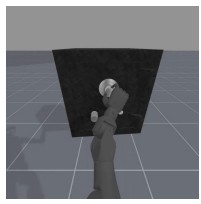 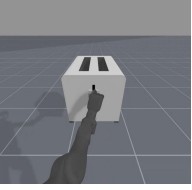 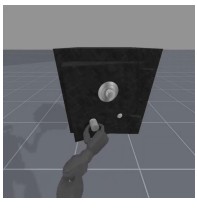 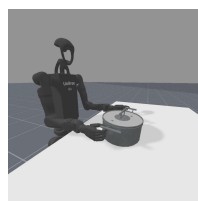 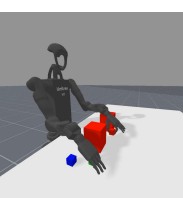

**Rotate Safe Knob**  **Press Toaster**  **Open Safe Door**  **Dual Lift Pot**  **Blocks Stack Hard With Barrier**

Figure 11: Illustration of the five additional challenging bimanual dexterous manipulation tasks used for supplementary evaluation: *rotate safe knob*, *press toaster*, *open safe door*, *dual lift pot*, and *blocks stack hard With barrier*. These tasks involve coordinated bimanual motions, contact-rich dexterous operations, and complex tabletop environments, highlighting the scalability of our framework.

**Experimental Setup.** The five additional tasks include *rotate safe knob*, *press toaster*, *open safe door*, *dual lift pot*, and *blocks stack hard with barrier*. In *rotate safe knob*, the robot precisely pinches and rotates a knob, requiring accurate fingertip control to prevent slippage. In *press toaster*, the robot presses a switch using a single finger, demonstrating fine dexterous motion beyond grasping or pinching. The *open safe door* task involves pulling open a door with a vertically oriented hinge, testing manipulation of articulated mechanisms. In *dual lift pot*, the robot must grasp both handles of a pot simultaneously and lift it, testing coordinated bimanual manipulation. Finally, *blocks stack hard with barrier* requires stacking blocks in an environment where more than 60% of the workspace is occupied by barriers, posing severe spatial constraints.

Table 7: Evaluation of additional challenging dexterous manipulation tasks. Each task is tested under randomized initial positions and orientations.

| Metric | Rotate Safe Knob | Press Toaster | Open Safe Door | Dual Lift Pot | Blocks Stack Hard With Barrier |
|---|---|---|---|---|---|
| **Randomization** | 5 cm × 5 cm, ±15° | 5 cm × 5 cm, ±15° | 5 cm × 5 cm, ±15° | 12 cm × 6 cm, ±15° | 6 cm × 8 cm, ±10° |
| **Success Rate(%)** | 91.09 | 87.72 | 95.24 | 93.00 | 73.00 |

**Experimental Results.** As shown in Tab. 7, our framework achieves high success rates across all five tasks, with three exceeding 90%. These results further validate the scalability of our method in handling bimanual coordination and contact-rich dexterous manipulation. Even in the most challenging case, *blocks stack hard with barrier*, where over 60% of the workspace is blocked by obstacles, the framework still achieves a 73% success rate, outperforming existing baselines under comparable conditions.

## C.4 Resource and Efficiency Analysis of HumanoidGen

We analyze average computational and token consumption across all 20 tasks in HGen-Bench, including both time cost and LLM resource usage. The detailed statistics are summarized in Tab. 8.

Table 8: Average time and resource consumption for each stage in the HumanoidGen data generation pipeline, computed over all 20 tasks in HGen-Bench. Manual annotation requires limited human effort, while all other stages are fully automated.

| Metric | Manual Annotation | Automatic Annotation | Scene Generation | Script Generation | Data Collection Execution |
|---|---|---|---|---|---|
| Time (s) | 4-6 | 1.17 | 12.64 | 18.31 | 14.40 |
| Resource Cost | Human Operator | Automated Process | LLM (2,958 tokens avg.) | LLM (3,745 tokens avg.) | Motion Planner (0.53 s) |

It is important to note that generating a new trajectory does not always require executing all stages of the pipeline. Depending on the scenario, certain steps can be skipped or reused, significantly improving efficiency: (i) For pose randomization of manipulated objects, tabletop obstacles, or room-level scene extensions, the system can directly execute the previously generated running scripts to collect new trajectories. The randomization operations are executed at the millisecond level and thus have negligible time cost. (ii) For tasks sharing identical tabletop setups but differing in manipulation procedures, such as *handover and storage* and *handover and storage cooperation*. For these tasks, scene initialization can be reused, requiring only the generation of new execution scripts. (iii) When different objects are used but all exist in the asset library, no manual intervention is required; automated scene generation and code synthesis are sufficient to produce new executable scripts for data collection. Only when introducing new object categories absent from the asset library is manual annotation needed, which takes merely 4–6 seconds per asset class.

These results demonstrate that HumanoidGen achieves high efficiency in both automated data generation and resource utilization, enabling scalable and reproducible large-scale evaluations in HGen-Bench.

## C.5 Comparison with Existing Generation Frameworks

To demonstrate the advantages of our proposed framework, we conduct a comparison with several state-of-the-art robot manipulation data generation frameworks, including both augmentation-based and zero-shot generation approaches.

**Augmentation-Based Frameworks.** These methods [51, 57] require an existing expert trajectory as the basis for data expansion, which typically involves teleoperation-based data collection. Moreover, manual annotation is needed to segment long-horizon demonstrations into multiple sub-stages before augmentation can be applied. When the task execution mode changes, such as switching from *left-to-right* to *right-to-left* handover, or modifying the stacking order in a block stacking task, new expert data collection and segmentation are needed. In contrast, our framework achieves object-level reusability and full automation without human intervention, enabling flexible and scalable task generation under varied manipulation settings.

Table 9: Comparison of different robot manipulation data generation frameworks. Only our proposed HumanoidGen provides a unified and fully automated pipeline that supports bimanual dexterous hands, scene generation, room-level synthesis, and dynamic collision management.

| Framework | Bimanual | Dexterous Hand | Tabletop Scene Generation | Room-level Scene Synthesis | Dynamic Collision Management |
|---|---|---|---|---|---|
| RoboGen [22] | Yes | No | Yes | Yes | No |
| Gensim [23] | No | No | Yes | No | No |
| Gensim2 [27] | No | No | Yes | Yes | No |
| RoboTwin [19] | Yes | No | No | No | No |
| RoboFactory [58] | Yes | No | No | No | No |
| HumanoidGen(Ours) | Yes | Yes | Yes | Yes | Yes |

**Zero-Shot Generation Frameworks.** For frameworks that support zero-shot data generation without relying on expert demonstrations, a comparison with HumanoidGen is summarized in Tab. 9. It can be clearly observed that HumanoidGen is the first framework to provide a systematic solution for bimanual dexterous hand manipulation problems. Other frameworks do not incorporate dexterous hands as end-effectors. RoboTwin [19] and RoboFactory [58] lack table-top scene generation and room-level scene scaling, both of which are included in HumanoidGen, demonstrating the comprehensiveness of our work. Additionally, only our framework includes dynamic collision management by LLMs, enhancing the flexibility and effectiveness of our framework in handling collisions during long-horizon tasks.

## C.6 Comparison Across Different Large Models

To further investigate the impact of different large models serving as the planner within our framework, we conduct a comparative study involving a reasoning model, a chat model, and a multimodal model, both with and without visual input. Specifically, we evaluate *DeepSeek-R1* (reasoning model), *DeepSeek-Chat-v3* (chat model), and *GPT-4o* in both its language-only and multimodal configurations. Four representative tasks, each from a distinct category in Fig. 5, are selected for this experiment.

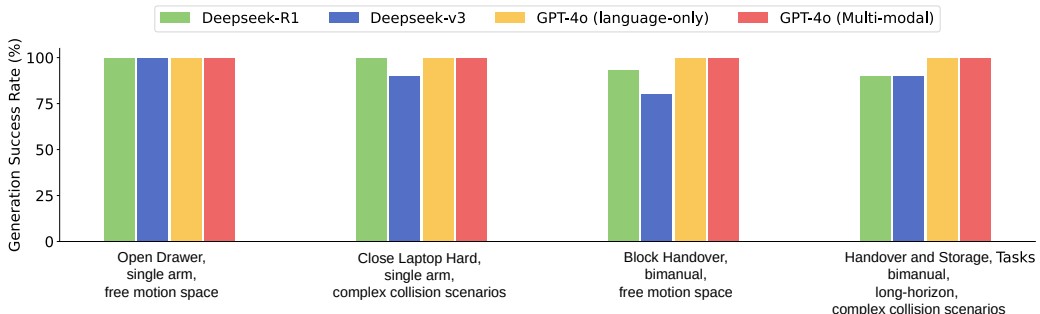

Figure 12: Comparison of different large models used as the planning module in our framework: a reasoning model (DeepSeek-R1), a chat model (DeepSeek-Chat-v3), and a multimodal model with and without visual input (GPT-4o language-only and GPT-4o multimodal).

As shown in Fig. 12, all four models achieve high success rates in generating valid constraints and executable planning code. Specifically, GPT-4o (with and without image inputs) shows a consistently good performance across all four tasks, whereas the Deepseek reasoning model and the Deepseek chat model have a slight performance decline when generating plans for bimanual and long-horizon tasks. The consistent high performance of these different models across various tasks demonstrates the effectiveness of our framework and its compatibility with diverse models.

## C.7 Task Descriptions in Data Generation Experiment

We describe our 20 tabletop-level tasks in detail in Tab. 10. The initial positions of target objects in all tasks are randomized. The tasks include 12 single-arm tasks and 8 bimanual tasks. Dexterous hands are used as end-effectors in all tasks. For bimanual tasks, the appropriate arms to manipulate target objects are selected according to the distance between the arms and the objects. Some tasks involve object handoffs between two hands, like *block handover* and *handover and storage*. Furthermore, *handover and storage cooperation* requires the coordination between both arms to complete the task.

## C.8 Task Descriptions in MCTS Experiment

As mentioned in §5.2, to evaluate the effectiveness of MCTS in enhancing the demonstration generation process of HumanoidGen, we simplified task descriptions to provide minimal guidance in the experiments. As shown in Tab. 11, we did not notice the operation order or the operation executor, both of which were inferred by the reasoning LLM.

# D  More Discussion on Limitations and Future Work

Despite the promising results achieved by our framework, several limitations remain. First, human intervention has not been completely eliminated. For asset categories and atomic manipulation types that have not been previously annotated, automatic labeling through annotation transfer is not possible, requiring manual annotation instead. Consequently, generating a task that involves a novel asset category still demands prior human annotation of its manipulation primitives. However, we note that several recent works have proposed learning-based methods for predicting contact points, axes, or grasp poses [59, 60, 61]. Although these approaches are not yet fully mature and still face generalization challenges, they demonstrate certain zero-shot capabilities that could be integrated into our annotation pipeline. Combining these advances with our generative framework holds promise for fully automated data generation, which we plan to explore in future work.

Second, our current framework cannot yet generate all types of dexterous manipulation tasks. On one hand, due to the limitations of the ManiSkill3 [52] physics engine, tasks involving deformable or fluid objects cannot be simulated. On the other hand, our arm control currently relies on a low-level motion planner, making it difficult to handle manipulation tasks with ambiguous or dynamic objectives, such as push-T or cloth flattening. These

Table 10: The descriptions of our 20 tasks.

| Task | Description |
|---|---|
| **Block Handover** | A rectangular cube is placed on the right side of the table. The right hand pinches the cube and moves it to hand over to the left hand. The left hand then pinches the cube and moves the cube above the target cube to release it. |
| **Blocks Stack Easy** | Two cubes, cube0 and cube1, are placed on the table. Initially, both the left and right hands simultaneously pinch cube0 and cube1, respectively. The left hand then put cube0 to a specified position. Following this, the right hand moves cube1 to put it above cube0. |
| **Blocks Stack Hard** | Three cubes, cube0, cube1, and cube2, are placed on the table. Initially, the left hand pinches cube0 and the right hand pinches cube1 simultaneously. The left hand puts cube0 at the target place. Then the right hand puts cube1 above cube0. Finally, the right hand picks up cube2, puts it above cube1. |
| **Close Box Easy** | A box is placed on the table. Initially, the right hand approaches the box lid from above. Then, the right hand moves to flip the box lid down. |
| **Close Box Hard** | A box is placed on the table. Initially, the right hand grasps the box lid. While maintaining its grasp, the right hand adjusts the box lid's openness to 0.3 and releases it. |
| **Close Drawer** | A drawer is placed on the table. Initially, the left hand moves to grasp the drawer handle. With the left hand attached to the drawer handle, the left hand pushes the drawer back to an openness of 0. Finally, the left hand releases the drawer handle. |
| **Close Laptop Easy** | A laptop is placed on the table. First, the right hand approaches the laptop screen from above. Then, the right hand moves to flip the laptop screen down. |
| **Close Laptop Hard** | A laptop is placed on the table. First, the right hand grasps the laptop screen from above. While maintaining its grasp, the right hand adjusts the laptop screen's openness to 0.3. Then, the right hand releases the laptop screen and moves above it. Finally, the right hand moves to flip the laptop screen fully down. |
| **Cup Pour Easy** | A cup and a bowl are placed on the table. First, the right hand grasps the cup and moves it above and in front of the bowl. Finally, the right hand tilts the cup to pour its contents into the bowl. |
| **Dual Bottles Pick Easy** | Two bottles, bottle0 and bottle1, are placed on the table. The left and right hands simultaneously grasp the two bottles and lift them to the target position. |
| **Dual Bottles Pick Hard** | Two bottles, bottle0 and bottle1, are placed on the table. The left and right hands simultaneously grasp the two bottles and lift them to the target position. |
| **Empty Cup Place** | A cup and a plate are placed on the table. The task is to use one hand to grasp the cup and place it directly over the plate. |
| **Handover and Storage** | A drawer and a rectangular cube0 are placed on the table. First, the left hand grasps the drawer handle and pulls it out to an openness of 0.9. Then the left hand releases the drawer handle. Next, the right hand pinches the cube and moves it to hand it over to the left hand. The left hand then moves to pitch the cube, taking it from the right hand. The left hand moves the cube above the open drawer and releases it to put it into the drawer. Afterwards, the left hand moves to grasp the drawer handle again, pushes it back to an openness of 0, and finally releases the drawer handle. |
| **Handover and Storage Cooperation** | A drawer and a rectangular cube0 are placed on the table. First, the left hand grasps the drawer handle, and the right hand pinches cube0 simultaneously. Then, the right hand moves the cube to an intermediate position while the left hand pulls the drawer out to an openness of 0.9 at the same time. The left hand then releases the drawer handle. Next, the left hand moves to pinch the cube, taking it from the right hand. The left hand moves the cube above the drawer and releases it to put it into the drawer. Afterwards, the left hand moves to grasp the drawer handle again, pushes it back to an openness of 0, and finally releases the drawer handle. |
| **Open Box Easy** | A box is placed on the table. First, the right hand moves to be under the box lid. Then the right hand moves to flip the box lid up. |
| **Open Box Hard** | A box is placed on the table. First, the right hand moves to grasp the box lid. While maintaining its grasp, the right hand adjusts the box lid's openness to 0.8 and releases it. After this, the right hand releases the box lid and moves to be under it. Finally, the right hand moves to flip the box lid up. |
| **Open Drawer** | A drawer is placed on the table. First, the left hand moves to grasp the drawer handle. With the left hand grasping the drawer handle, it pulls the drawer out to an openness of 1. Finally, the left hand releases the drawer handle. |
| **Open Laptop Easy** | A laptop is placed on the table. First, the right hand moves to be under the laptop screen. Then, the right hand moves to flip the laptop screen up. |
| **Open Laptop Hard** | A laptop is placed on the table. First, the right hand grasps the laptop and flips it up to an openness of 0.55. Then, the right hand releases the initial grasp and moves to be under the laptop screen. Finally, it moves to flip the laptop screen to be fully up. |
| **Pyramid Stack** | Three cubes, cube0, cube1, and cube2, are placed on the table. First, the left hand and the right hand simultaneously pinch cube0 and cube1, respectively. The left hand then moves cube0 to put it at the target position. After this, the right hand moves cube1 to put it on the right side of cube0. Then, the right hand pinches cube2 and moves it to put it in the middle and above cube0 and cube1. |

tasks involve continuous state adjustments rather than planning toward a single target pose, which cannot be realized solely through motion planning. In the future, we plan to extend our framework to support more diverse physics engines, such as IsaacSim [15] and MuJoCo [16], enabling richer simulation scenarios. Moreover, leveraging the extensibility of our framework, we will explore combining goal-conditioned motion planning with goal-agnostic pre-trained atomic operation models, forming a unified library of atomic operations to support a wider range of dexterous manipulation tasks.

Finally, we plan to expand HGen-Bench with additional assets, scenes, tasks, and evaluation policies to enable more comprehensive benchmarking and broader applicability.

Table 11: The descriptions of 4 tasks in the MCTS experiment.

| Task | Description |
|---|---|
| Block Stack Single | Stack cube1 on top of cube0. |
| Blocks Stack Easy | Stack the two cubes on the table into a single pile. |
| Blocks Stack Hard | Stack the three cubes on the table into a single pile. |
| Pyramid Stack | Stack the three cubes into a pyramid shape in the center area, with two cubes at the bottom and one cube stacked on top of them. |

# E  Prompts and Generation Samples

We show the prompt templates and the sample code generated by LLMs in scene generation and demonstration script generation (with and without integrating MCTS). The meanings of the placeholders in the prompt templates are explained in Table 12.

Table 12: The placeholders in the prompt templates and their meanings.

| Placeholder | Meaning |
|---|---|
| ASSET_INFO | The information of the assets in the asset library. |
| ASSETS_ATTRIBUTES | The default attributes of the assets. |
| INITIAL_ASSET_STATE | The asset states when the scene is initialized, before any actions are executed. |
| CURRENT_ASSET_STATE | The current states of the assets on the tabletop. |
| EXECUTED_CODE | Code for actions executed by the robot to transform the asset state on the tabletop from the initial state to the current state. |
| PROHIBITED_ACTION | The actions prohibited for the next step. |
| ASSETS_STATUS | The current states of the assets in the task scene. |
| ROBOT_END_EFFECTOR | The current state of the robot end-effector. |

## E.1  Prompt Template for Scene Generation

```
1 You are a professional AI simulation environment code generation assistant capable
      of generating logical reasoning and accurate code. You need to generate
      reasoning monologues for the initial scene of the task based on the user's
      given task, i.e., what kind of initial scene to generate and why the initial
      scene is designed this way. Afterward, provide an answer that includes the
      code to construct the scene.
2 ======
3 Scene information (all coordinates are in the world coordinate system):
4 Dual-arm robot, pose.p=[-0.85,0,0], pose.q=[1,0,0,0]
5 Table surface,  "x from -0.42 to -0.19, y from -1.1 to 1.16, z=0"
6 ======
7 Available assets:
8 ASSETS_INFO
9 ======
10 Code Example: Task to place two cubes side by side.
11 ```python
12 from humanoidgen.envs.example.task_env import * # Import necessary libraries
13 @register_env("place_cubes_side_by_side", max_episode_steps=200) # Register the
      environment, name can be set based on the task
14 class PutTwoCubeAdjacentEnv(TableSetting): # Must inherit from TableSetting
15     env_name= "place_cubes_side_by_side"
16     def _load_scene(self, options: Dict):      # Load objects
17         super()._load_scene(options)
18         self._add_object(type_name="cube", type_id=1)  # name="cube", obj_id=0,
              yellow cube
19         self._add_object(type_name="cube", type_id=0)  # name="cube", obj_id=1,
              green cube
20
21     def _initialize_episode(self, env_idx: torch.Tensor, options: Dict):  # Set
          object positions
```

```
22          super()._initialize_episode(env_idx, options)
23          self._set_object_pose(type_name="cube", obj_id=0, pose=sapien.Pose(
24              p=[-0.38, 0.28, 0.02],
25              q=[1, 0, 0, 0]
26          ))
27          self._set_object_pose(type_name="cube", obj_id=1, pose=sapien.Pose(
28              p=[-0.32, -0.32, 0.02],
29              q=[1, 0, 0, 0]
30          ))
31
32      def check_success(self):
33          print("=========== check_success ===========")
34          p0 = self.cube[0].pose.p.numpy()[0]
35          p1 = self.cube[1].pose.p.numpy()[0]
36          target_position_0=[-0.3, 0.02,0.02]
37          target_position_1=[-0.3, -0.02,0.02]
38          eps = [0.03, 0.03]
39          success_0 = all(abs(p0[i] - target_position_0[i]) <= eps[i] for i in range
                  (2))
40          success_1 = all(abs(p1[i] - target_position_1[i]) <= eps[i] for i in range
                  (2))
41          print("cube[0] position:", p0)
42          print("cube[0] target position:", target_position_0)
43          print("cube[1] position:", p1)
44          print("cube[1] target position:", target_position_1)
45          print("success_0:", success_0)
46          print("success_1:", success_1)
47          return success_0 and success_1
48  '''
49  ======
50  Notes:
51  - `type_id` represents different models of the same type of object, while `obj_id`
        indicates the order in which objects of the same type are added.
52  - The left hand's operable range is  x  [-0.42, -0.19],  y  [-0.07, 0.36]; the
        right hand's range is  x  [-0.42, -0.19],  y  [-0.36, 0.07]. Objects must be
        placed within these intervals (note: meeting the range is a necessary but not
         sufficient condition    successful grasping may still fail due to object
        orientation, collisions, or gripper constraints).
53  - The robot's coordinate system aligns with the world frame: +X (front of robot),
        +Y (left of robot), +Z (up of robot).
54  - The object's bounding box (BBX) represents its dimensions in its own coordinate
        system (axis-aligned when the orientation quaternion is q=[1,0,0,0]). For
        example, a can with BBX [0.054, 0.1, 0.054] has lengths of 0.054 units along
        X/Z-axes and 0.1 units along the Y-axis.
55  ======
56  Now please generate the initial scene of task "TASKNAME".
```

## E.2    Sample Scene Code for Blocks Stack Hard Generated by HumanoidGen

```
1  from humanoidgen.envs.example.task_env import *
2
3  @register_env("blocks_stack_hard", max_episode_steps=200)
4  class BlocksStackHardMctsEnv(TableSetting):
5      env_name= "blocks_stack_hard"
6
7      def _load_scene(self, options: Dict):
8          super()._load_scene(options)
9          self._add_object(type_name="cube", type_id=0)
10         self._add_object(type_name="cube", type_id=1)
11         self._add_object(type_name="cube", type_id=2)
12
13     def _initialize_episode(self, env_idx: torch.Tensor, options: Dict):
14         super()._initialize_episode(env_idx, options)
15         default_pose = [
16             sapien.Pose(p=[-0.32, 0.32, 0.05], q=[1,0, 0, 0]),
17             sapien.Pose(p=[-0.32, -0.32, 0.05], q=[1, 0, 0, 0]),
18             sapien.Pose(p=[-0.45, -0.32, 0.05], q=[1, 0, 0, 0]),
19         ]
20         if self.random_scene:
21             default_pose=self.get_random_pose(default_pose=default_pose)
22             if not self.random_once or (self.random_once and not hasattr(self, "
                    random_pose")):
23                 self.random_pose = default_pose
24         self._set_object_pose(
```

```
25              type_name="cube",
26              obj_id=0,
27              pose=self.random_pose[0]
28          )
29
30          self._set_object_pose(
31              type_name="cube",
32              obj_id=1,
33              pose=self.random_pose[1]
34          )
35
36          self._set_object_pose(
37              type_name="cube",
38              obj_id=2,
39              pose=self.random_pose[2]
40          )
41
42      def check_success(self):
43          print("=========== check_success ===========")
44          positions = [
45              self.cube[0].pose.p.numpy()[0],
46              self.cube[1].pose.p.numpy()[0],
47              self.cube[2].pose.p.numpy()[0]
48          ]
49          positions.sort(key=lambda p: p[2])
50          eps_xy = 0.03
51          eps_z = 0.005
52          height_diff = 0.04
53          xy_aligned = (
54              abs(positions[0][0] - positions[1][0]) <= eps_xy and abs(positions
                  [0][1] - positions[1][1]) <= eps_xy and
55              abs(positions[1][0] - positions[2][0]) <= eps_xy and abs(positions
                  [1][1] - positions[2][1]) <= eps_xy
56          )
57          z_aligned = (
58              abs(positions[1][2] - positions[0][2] - height_diff) <= eps_z and
59              abs(positions[2][2] - positions[1][2] - height_diff) <= eps_z
60          )
61          return xy_aligned and z_aligned
```

### E.3 Prompt Template for Demonstration Script Generation

```
1 You are a professional assistant for generating code that enables dual-arm robots
      to perform tabletop tasks. Please generate logically structured code to
      execute the user-specified tasks based on the provided context.
2 ======
3 Scene information:
4 Dual-arm robot, pose.p=[-0.85,0,0], pose.q=[1,0,0,0], "robot_base_link" is the
      connection between the "torso_link" and the "pelvis".
5 Table surface, x [-0.42, -0.19], y [-1.1, 1.16], z=0
6 ======
7 Assets attributes (The default state of the assets, not the current state):
8 ASSETS_ATTRIBUTES
9 ======
10 Assets status (The current state of the assets):
11 ASSETS_STATUS
12 ======
13 Robot end-effector (wrist) current status:
14 ROBOT_END_EFFECTOR
15 ======
16 Available functions:
17 def hand_pre_grasp(hand_name):
18     Function: Adjusts the thumb movement of the specified hand to position it
          opposite the index finger, preparing for subsequent grasping operations.
          This should typically be called before executing 'move_to_pose_with_screw
          ' to reach the pre-grasp pose.
19     Return: None
20     Args:
21     - hand_name (str, optional): "all" (both), "right", or "left". Default: "all".
22     Example:
23         planner.hand_pre_grasp("left")  # Control the left hand fingers to assume
              the pre-grasp pose
24         planner.hand_pre_grasp("right") # Control the right hand fingers to assume
              the pre-grasp pose
```

```
25          planner.hand_pre_grasp("all")
26
27  def hand_grasp(hand_name, grasp_object, obj_id):
28      Function: Control the hand to close.
29      Return: None
30      Args:
31      - hand_name (str, Mandatory): "right" or "left".
32      - grasp_object (str, Mandatory): The type of the grasped object.
33      - obj_id (int, Mandatory): The object id of the grasped object.
34      Example:
35          planner.hand_grasp("left",grasp_object="can",obj_id=0)   # Control the
                left hand close to grasp the bottle 0
36          planner.hand_grasp("right",grasp_object="can",obj_id=1)  # Control the
                right hand close to grasp the bottle 0
37
38  def hand_pre_pinch(hand_name):
39      Usage: Similar to the usage of the hand_pre_grasp function.
40
41  def hand_pinch(hand_name, pinch_object, obj_id):
42      Usage: Similar to the usage of the hand_grasp function.
43
44  def open_hand(self, hand_name):
45      Function: Control the hand to open the fingers.
46      Return: None
47      Args:
48      - hand_name (str, Mandatory): "right" or "left".
49
50  def generate_constraints(self,obj_name,obj_id,action,hand_name):
51      Function: Generate the constraints for the end-effector pose when performing a
            specific action.
52      Return: constraints
53      Args:
54      - obj_name (str, Mandatory): The object to be operated.
55      - obj_id (int, Mandatory): The object id of the operated object.
56      - action (str, Mandatory): The action name corresponding to the constraints.
            Can be "pinch", "grasp", "target", "move", "target1", "target2"
57          - "pinch": Move to the pose for pinching the object.
58          - "grasp": Move to the pose for grasping the object.
59          - "target": Move the "target" pose
60          - "target1": Move the "target1" pose
61          - "target2": Move the "target2" pose
62          - "move": Move to a specific pose relative to an object.
63      - hand_name (str, Mandatory): "right" or "left".
64      - openness (float, Optional): [necessary for openness constraint generation]
            the openness of the manipulated object after being manipulated. The value
             should be between 0 or 1. This argment is only used when the action is "
            grasp" or "pinch".
65      - relative_obj_name (str, Optional): [necessary for relative pose constraint
            generation] the name of the object to be used as a reference for defining
             the relative pose.
66      - relative_obj_id (int, Optional): [necessary for relative pose constraint
            generation] the id of the object to be used as a reference for defining
            the relative pose.
67      - relative_p (np.array, Optional): [necessary for relative pose constraint
            generation] the relative position of the end-effector to the reference
            object.
68      Example:
69          constraint_l=planner.generate_constraints(obj_name="can", obj_id=0, action
                ="pinch", hand_name="left")  # Generate the constraints for the left
                hand end-effector to move to the pre-pinch pose for pinching can0
70          constraint_l=planner.generate_constraints(obj_name="can", obj_id=0, action
                ="grasp", hand_name="left")  # Generate the constraints for the left
                hand end-effector to move to the pre-grasp pose for grasping can0
71          constraint_l=planner.generate_constraints(obj_name="can", obj_id=0, action
                ="target",  hand_name="left")  # Generate the constraints for the
                left hand end-effector to move to the target pose after grasping can0
72          constraint_r=planner.generate_constraints(obj_name="can", obj_id=1, action
                ="move", hand_name="right",relative_obj_name="can",relative_obj_id=0,
                relative_p=np.array([0,0,0.06]))   # Generate the end-effector pose
                for the right hand after grasping can1 to place it 6cm above can0
                along the world coordinate system's z-axis
73          constraint_l=planner.generate_constraints(obj_name="
                some_articulated_object", obj_id=0, action="target1",  hand_name="
                left")  # Generate the constraints for the left hand end-effector to
                move to the target1 pose with the left hand pushing the object "
                some_articulated_object" to move along its articulation axis
74          constraint_r=planner.generate_constraints(obj_name="
                some_articulated_object", obj_id=0, action="target2",  hand_name="
```

```
                 right")  # Generate the constraints for the right hand end-effector
                 to move to the target2 pose with the right hand pushing the object "
                 some_articulated_object" to move along its articulation axis
75          constraint_l=planner.generate_constraints(obj_name="
                 some_articulated_object", obj_id=0, action="grasp",  hand_name="left"
                 )  # Generate the constraints for the left hand end-effector to move
                 to the pre-grasp pose for grasping the object "
                 some_articulated_object"
76          constraint_l=planner.generate_constraints(obj_name="
                 some_articulated_object", obj_id=0, action="grasp",  hand_name="left"
                 , openness=0.7)  # Generate the constraints for the left end-effector
                  pose after grasping the articulated_obj. This constraint is used for
                  planning a motion to turn the "some_articulated_object" to an
                  openness of 0.7.

78  def generate_end_effector_pose(constraints,hand_name):
79      Function: Calculate the end-effector pose that satisfies the given constraints
                 .
80      Return: _,target_effector_pose
81      Args:
82      - constraints (list, Mandatory)
83      - hand_name (str, Mandatory): "left" or "right"
84      Example:
85          _, target_effector_pose_l = planner.generate_end_effector_pose(
                 constraint_l,hand_name="left")

87  def move_to_pose_with_screw(pose: sapien.Pose, hand_name, attach_obj=False,
         object_name=None, object_id=0):
88      Function: Control the robotic arm to move the end-effector of 'hand_name' to
             the 'pose'.
89      Return: None
90      Args:
91      - pose (sapien.Pose or list, Mandatory): The target pose for the movement of '
             hand_name'.
92      - hand_name (str, Mandatory): "all", "left" or "right".
93      - attach_obj (bool or list, Mandatory): Whether there is an attached object in
              the hand during the movement.
94      - object_name (str or list): [Required when 'attach_obj' is True] The type of
              the attached object.
95      - object_id (int or list, Mandatory): [Required when 'attach_obj' is True] The
              object id of the attached object.
96      Example:
97          # If 'hand_name' is 'all', ensure that other parameters are of list type,
                 where index [0] and [1] correspond to the parameters for the left and
                  right hands, respectively.
98          # you need to indicate the objects that are currently in hand (grasped or
                 pinched) with args attach_obj, object_name, and object_id. If both
                 hands are holding objects, set these args with list.
99          planner.move_to_pose_with_screw(target_effector_pose,"all",attach_obj=[
                 False,False])
100         planner.move_to_pose_with_screw(planner.right_hand_init_pose,hand_name="
                 right") # The right hand returns to the initial pose.

102 ======
103 Code example:
104 1. The right hand grasps the can. Initial conditions: Not yet performed any action

106 step 0: Set the right hand fingers to a pre-grasp pose
107 ```python
108     # Set the right hand fingers to a pre-grasp pose
109     planner.hand_pre_grasp("right")
110 ```

112 step 1: Move the right hand to grasp pose
113 ```python
114     constraint_r=planner.generate_constraints(obj_name="can", obj_id=0, action="
             grasp", hand_name="right")
115     # Generate the target pose for the end-effector and move the right hand to the
              target pose
116     _, target_effector_pose = planner.generate_end_effector_pose(constraint_r,
             hand_name="right")
117     planner.move_to_pose_with_screw(target_effector_pose,"right",attach_obj=False)
118 ```

120 step 2: Close the right hand to grasp the can
121 ```python
122     # Close the right hand to grasp the can
123     planner.hand_grasp("right",grasp_object="can",obj_id=0)
```

```
124 ```
125
126 ======
127 Notes:
128 - All coordinate information, unless otherwise specified, is in the world
         coordinate system.
129 - The robot's coordinate system aligns with the world frame: +X (front of robot),
         +Y (left of robot), +Z (up of robot).
130 - `type_id` represents different models of the same type of object, while `obj_id`
         indicates the order in which objects of the same type are added.
131 - The object's bounding box (bbox) represents its dimensions in its own coordinate
         system (axis-aligned when the orientation quaternion is q=[1,0,0,0]). For
         example, a can with bbox:{"min": [-0.08,-0.08,-0.02],"max": [0.08,0.08,0.02]}
         has lengths of 0.16 units along X/Y-axes and 0.04 units along the Z-axis.
132 - Please place each step in a separate code block, enclosed within ```python ```
         tags. The number of steps should correspond exactly to the number of ```
         python ``` blocks in your response, with no extra or missing blocks.
133 - The parameters in the code block corresponding to each step are not shared.
         Please do not carry over the parameters set in the previous step to the next
         step.
134 - When selecting the hand to operate on an object, please choose the hand that is
         closer to the object.
135 - Each line of code must have at least one level of indentation.
136 - - The left hand's operable range is  x  [-0.42, -0.19],  y  [-0.07, 0.36]; the
         right hand's range is  x  [-0.42, -0.19],  y  [-0.36, 0.07]. Objects must be
         placed within these intervals (note: meeting the range is a necessary but not
          sufficient condition    successful grasping may still fail due to object
         orientation, collisions, or gripper constraints).
137 ======
138 Please generate the code to execute 'TASK_DESCRIPTION'.
```

## E.4 Sample Demonstration Code for Open Drawer Generated by HumanoidGen

```python
from humanoidgen.motion_planning.h1_2.solution.generated.solver_env import *
def step(planner:HumanoidMotionPlanner):
    # Step 1: Left hand grasps the drawer
    planner.hand_pre_grasp("left")
    constraint_r = planner.generate_constraints(
            obj_name="drawer",
            obj_id=0,
            action="grasp",
            hand_name="left"
        )
    _, target_effector_pose = planner.generate_end_effector_pose(constraint_r, "
        left")
    planner.move_to_pose_with_screw(
        target_effector_pose,
        "left",
        attach_obj=False
    )
    planner.hand_grasp("left", grasp_object="drawer", obj_id=0)
    # Step 2: Left hand opens the drawer to an openness of 1
    constraint_open = planner.generate_constraints(
        obj_name="drawer",
        obj_id=0,
        action="grasp",
        hand_name="left",
        openness=1
    )
    _, target_open_pose = planner.generate_end_effector_pose(constraint_open, "
        left")
    planner.move_to_pose_with_screw(
        target_open_pose,
        "left",
        attach_obj=True,
        object_name="drawer",
        object_id=0
    )
    # Step 3: Left hand releases the drawer
    planner.open_hand("left")
```

## E.5 Prompt Template for MCTS-integrated Demonstration Script Generation

```
1  You are a professional assistant for generating code that enables dual-arm robots
       to perform tabletop tasks. Please generate logically structured code to
       execute the user-specified tasks based on the provided context.
2  ======
3  Static Scene Elements:
4  The static scene elements include a robot and a table. The robot's base_link (
       position between torso_link and pelvis) has a pose of pose.p = [-0.85, 0, 0],
        pose.q = [1, 0, 0, 0]. The table surface spans x     [-0.42, -0.19], y
       [-1.1, 1.16], z = 0.
5  ======
6  World Coordinate System Information:
7  Since the robot's base coordinate frame has pose.q = [1, 0, 0, 0], its xyz
       directions align with the world coordinate system: x points forward, y points
        to the robot's left, and z points upward.
8  ======
9  Intrinsic Asset Attributes:
10 ASSETS_ATTRIBUTES
11 Note: The 'orientation' in 'status' represents the rotation in the world
       coordinate system under its corresponding 'description'.
12 ======
13 Initial Asset State:
14 INITIAL_ASSET_STATE
15 Note: The asset(object) state(pose) when the scene is just initialized, before any
        actions are executed. Pose consists of Cartesian coordinates and a
       quaternion in the world coordinate system: [x, y, z, w, x, y, z](The
       following 7-dimensional poses all follow this format).
16 ======
17 Current Asset State:
18 CURRENT_ASSET_STATE
19 Note: Current states(poses) of assets(objects) on the tabletop
20 ======
21 Executed Action Code:
22 EXECUTED_CODE
23 Note:
24 - Code for actions executed by the robot to transform the asset state on the
       tabletop from the initial state to the current state.
25 ======
26 Prohibited Next Actions:
27 PROHIBITED_ACTION
28 Note: The next step cannot execute the prohibited actions listed above. 'Same'
       means completely identical. For 'move', any addition, removal, or
       modification of constraints is considered a different action. Applies only to
        the next step; subsequent steps can still choose those actions.
29 ======
30 Robot Attributes:
31 hand_key_point:{
32     "left_hand":["base_left_hand","grasp_point_base_left_hand","
           pinch_point_base_left_hand"],
33     "right_hand":["base_right_hand","grasp_point_base_right_hand","
           pinch_point_base_right_hand"]
34 }
35 hand_key_axis:{
36     "left_hand":["left_pinch_axis","left_pinch_wrist_2_palm_axis","
           left_ring_2_index","left_grasp_axis","left_grasp_wrist_2_palm_axis"],
37     "right_hand":["right_pinch_axis","right_pinch_wrist_2_palm_axis","
           right_ring_2_index","right_grasp_axis","right_grasp_wrist_2_palm_axis"]
38 }
39 default_pose:{
40     "left_hand":[left_hand_init_pose],
41     "right_hand":[right_hand_init_pose]
42 }
43 ======
44 Available Class:
45 Constraint(env,type,end_effector_frame,hand_key_point,object_key_point,hand_axis,
       object_axis):
46     Function:
47         "Constraint" defines hard constraints that must be strictly satisfied.
48         Establishes spatial equivalence constraints between:
49             - (point2point) Specified end-effector point     target point.
50             - (parallel) Specified end-effector axis direction     target axis
                   direction.
51     Args:
52     - env (Environment, Mandatory): The planner's bound operating environment.
           Must reference the planner's environment instance through `planner.env`
           property.
```

```python
53        - type (str, Mandatory): "point2point" or "parallel".
54        - end_effector_frame (str, Mandatory): "l_hand_base_link" or "r_hand_base_link
              ".
55        - hand_key_point (np.ndarray): [Required when 'type' is 'point2point'] Pre-
              motion end-effector anchor point in world coordinates.
56        - object_key_point (np.ndarray): [Required when 'type' is 'point2point']
              Target's corresponding point in world coordinates.
57        - hand_axis (np.ndarray): [Required when 'type' is 'parallel'] Pre-motion end-
              effector alignment axis in world coordinates.
58        - object_axis (np.ndarray): [Required when 'type' is 'parallel'] Target's
              reference axis in world coordinates.
59      Example:
60          # Right palm facing down to grasp(Top-down grasping of the object)
61          Constraint(
62              env=planner.env,
63              type="parallel",
64              end_effector_frame="r_hand_base_link",
65              hand_axis=get_axis_in_env(planner.env,axis_name="right_grasp_axis"), #
                      The back of the palm points toward the front of the palm. Pinch
                      action is similar.
66              object_axis=np.array([0,0,-1]), # World frame,
67          )
68
69          # Right palm facing up to grasp(Down-Top grasping of the object). If there
                   is a table or other objects below the target, it often causes
                   collisions and makes it unreachable.
70          Constraint(
71              env=planner.env,
72              type="parallel",
73              end_effector_frame="r_hand_base_link",
74              hand_axis=get_axis_in_env(planner.env,axis_name="right_grasp_axis"), #
                      The back of the palm points toward the front of the palm. Pinch
                      action is similar.
75              object_axis=np.array([0,0,1]), # World frame,
76          )
77
78          # The direction from the right wrist to the palm center is facing forward.
79          Constraint(
80              env=planner.env,
81              type="parallel",
82              end_effector_frame="r_hand_base_link",
83              hand_axis=get_axis_in_env(planner.env,axis_name="
                      right_grasp_wrist_2_palm_axis"), # The direction from the wrist
                      to the palm center
84              object_axis=np.array([1,0,0]), # World frame
85          )
86
87          # The direction from the right pinky finger to the index finger is
                   parallel to the x-axis of object0.
88          Constraint(
89              env=planner.env,
90              type="parallel",
91              end_effector_frame="r_hand_base_link",
92              hand_axis=get_axis_in_env(planner.env,axis_name="right_ring_2_index"),
                      # The direction from the right pinky finger to the index finger
93              object_axis=get_axis_in_env(planner.env, "x", obj_type="object",
                      obj_id=0), # The coordinates of object0's x-axis in the world
                      coordinate system.
94          )
95
96          # After object0 is in hand, align its x-axis parallel to the world
                   coordinate system's z-axis, making its x-axis point upward.
97          Constraint(
98              env=planner.env,
99              type="point2point",
100             end_effector_frame="r_hand_base_link",
101             hand_key_point=get_axis_in_env(planner.env, "x", obj_type="object",
                      obj_id=0),
102             object_key_point=object_axis=np.array([0,0,1]), # np.array([0, 0.1,
                      0.1]) in the object0's coordinate system.
103         )
104
105         # The right-hand grasp point coincides with [0, 0.1, 0.1] in the object0's
                   coordinate system.
106         Constraint(
107             env=planner.env,
108             type="point2point",
109             end_effector_frame="r_hand_base_link",
```

```
110              hand_key_point=get_point_in_env(planner.env,point_name="
                     grasp_point_base_right_hand"),
111              object_key_point=get_point_in_env(planner.env,type_name="object",
                     obj_id=0,related_point=np.array([0, 0.1, 0.1])), # np.array([0,
                     0.1, 0.1]) in the object0's coordinate system.
112          )
113
114          # After object0 is in hand, position it 0.1m above object1.
115          Constraint(
116              env=planner.env,
117              type="point2point",
118              end_effector_frame="r_hand_base_link",
119              hand_key_point=get_point_in_env(planner.env,type_name="object",obj_id
                     =0),
120              object_key_point=get_point_in_env(planner.env,type_name="object",
                     obj_id=1)+np.array([0, 0, 0.1]), # np.array([0, 0, 0.1]) in the
                     world(robot) coordinate system.
121          )
122      Note:
123          'hand_key_point' and 'object_key_point' represent two points in the world
                 coordinate system. 'hand_key_point' moves with the corresponding end-
                 effector, and type='point2point' indicates the intention for the two
                 points to coincide.
124          'hand_axis' and 'object_axis' have similar meanings to 'point'.
125          'right_grasp_axis' or 'right_pinch_axis' parallel [0,0,-1] means executing
                 with the palm facing downward.
126          'right_grasp_axis' or 'right_pinch_axis' parallel [1,0,0] means executing
                 with the palm facing forward.
127          'right_grasp_axis' or 'right_pinch_axis' parallel [0,1,0] means executing
                 with the palm facing left.
128          'left_grasp_axis' or 'left_pinch_axis' parallel [0,-1,0] means executing
                 with the palm facing right.
129
130
131 Const(env,type,end_effector_frame,hand_key_point,object_key_point,hand_axis,
        object_axis):
132     The usage is similar to the Constraint class, but it is not a strict
            constraint; it is an optimization objective.
133 ======
134 Available functions:
135
136 def get_point_in_env(env, point_name,type_name, obj_id,related_point,openness):
137     Function: Get specified point position in world frame.
138     Return: point_position (np.ndarray)
139     Args:
140     - env (Environment, Mandatory): The planner's bound operating environment.
            Must reference the planner's environment instance through `planner.env`
            property.
141     - point_name (str, optional): The point name. Example: "
            grasp_point_base_left_hand".
142     - type_name (str, optional): The reference object relative to the target point
            .
143     - obj_id (int, optional): The reference object id.
144     - related_point (np.ndarray, optional): The coordinates of the target point
            relative to the object center in the object coordinate system. Default:
            np.array([0,0,0])
145     - openness (int, optional): Represents the openness degree of the articulated
            object. If type_name is an articulated object, this value can be set to
            obtain the coordinates corresponding to point_name at a specific openness
             degree.
146     Example:
147         # Get the current coordinates of the right-hand pinch point in the world
                coordinate system.
148         get_point_in_env(planner.env,point_name="grasp_point_base_pinch_hand")
149
150 def get_axis_in_env(env, axis_name, obj_type, obj_id):
151     Function: Get specified direction vector in world frame.
152     Return: direction_vector (np.ndarray)
153     Args:
154     - env (Environment, Mandatory): The planner's bound operating environment.
            Must reference the planner's environment instance through `planner.env`
            property.
155     - axis_name (str, Mandatory): The axis name. Example: "right_pinch_axis".
156     - obj_type (str, optional): The reference object relative to the target axis.
157     - obj_id (int, optional): The reference object ID.
158
159 def open_hand(self, hand_name):
160     Function: Control the hand to open.
```

```
161      Return: None
162      Args:
163      - hand_name (str, Mandatory): "all", "right", or "left".
164
165  def hand_pre_grasp(hand_name):
166      Function: Adjusts the thumb movement of the specified hand to position it
              opposite the index finger, preparing for subsequent grasping operations.
              This should typically be called before executing 'move_to_pose_with_screw
              ' to reach the pre-grasp pose.
167      Return: None
168      Args:
169      - hand_name (str, Mandatory): "all", "right", or "left". Default: "all".
170
171  def hand_grasp(hand_name, grasp_object, obj_id):
172      Function: Control the hand to close.
173      Return: None
174      Args:
175      - hand_name (str, Mandatory): "right", or "left".
176      - grasp_object (str, Mandatory): The type of the grasped object. Example: "can
              ".
177      - obj_id (int, Mandatory): The object ID of the grasped object. Example: If
              grasp_object="can" and obj_id=0, it indicates the first can object added
              to the environment.
178
179  def hand_pre_pinch(hand_name):
180      Function: Adjusts the thumb movement of the specified hand to position it
              opposite the index finger, preparing for subsequent pinching operations.
              This should typically be called before executing 'move_to_pose_with_screw
              ' to reach the pre-pinch pose.
181      Return: None
182      Args:
183      - hand_name (str, optional): "all", "right", or "left". Default: "all".
184
185  def hand_pinch(hand_name, pinch_object, obj_id):
186      Function: Only control the closing operation of the thumb and index finger.
              Typically used for small and hard-to-grasp objects.
187      Return: None
188      Args:
189      - hand_name (str, Mandatory): "right", or "left".
190      - pinch_object (str, Mandatory): The type of the pinched object. Example: "can
              ".
191      - obj_id (int, Mandatory): The object ID of the pinched object. Example: If
              pinch_object="can" and obj_id=0, it indicates the first can object added
              to the environment.
192
193   def generate_end_effector_pose(constraints,hand_name):
194      Function: Calculate the target pose that the robotic arm needs to move to in
              this step based on the constraint and cost.
195      Return: _,target_effector_pose
196      Args:
197      - constraints (list, Mandatory): A list of Constraint objects, with the number
               of Constraints determined by specific requirements.
198      - hand_name (str, Mandatory): "left" or "right"
199
200  def move_to_pose_with_screw(pose: sapien.Pose, hand_name, attach_obj=False,
          object_name=None, object_id=0):
201      Function: Control the robotic arm to move the end-effector of 'hand_name' to
              the 'pose'.
202      Return: None
203      Args:
204      - pose (sapien.Pose, Mandatory): The target pose for the movement of '
              hand_name'.
205      - hand_name (str, Mandatory): The specified robotic arm for the movement.
206      - attach_obj (bool, Mandatory): Whether there is an attached object in the
              hand during the movement.
207      - object_name (str): [Required when 'attach_obj' is True] The type of the
              attached object.
208      - obj_id (int, Mandatory): [Required when 'attach_obj' is True] The object id
              of the attached object.
209
210  ======
211  Incorrect example code:
212  1. The right hand grasps the can when cthe an is on the table:
213  step 1: Grasp can0
214  ```python
215      # Set the right-hand fingers to a pre-grasp pose
216      planner.hand_pre_grasp("right")
217
```

```
218      constraints=[]
219
220      constraints.append(
221          Constraint(
222              env=planner.env,
223              type="parallel",
224              end_effector_frame="r_hand_base_link",
225              hand_axis=get_axis_in_env(planner.env, axis_name="right_grasp_axis"),
226              object_axis=np.array([0, 0, 1])
227          )
228      )
229      # Generate the target pose for the end-effector and move the right hand to the
             target pose
230      _, target_effector_pose = planner.generate_end_effector_pose(constraints,
             hand_name="right")
231      planner.move_to_pose_with_screw(target_effector_pose,"right",attach_obj=False)
232
233      # Close the right hand to grasp the can
234      planner.hand_grasp("right",grasp_object="can",obj_id=0)
235  ```
236  Reason for the error: `object_axis=np.array([0, 0, 1])` represents the positive z-
         axis in the world coordinate system. If it is parallel to the grasp axis, it
         indicates a bottom-up grasping direction. For an object on a table, the hand
         cannot reach below the object. To perform a top-down grasp along the -z
         direction, use `object_axis=np.array([0, 0, -1])`.
237
238  ======
239  Correct example code:
240  1. The right hand grasps the can from right to left, then lifts it by 0.1m, and
         then releases it.
241  step 1: Grasp can0
242  ```python
243      # Set the right-hand fingers to a pre-grasp pose
244      planner.hand_pre_grasp("right")
245
246      constraints=[]
247      # Add a point-to-point constraint to align the grasp point of the right hand
             with the can's center point
248      constraints.append(
249          Constraint(
250              env=planner.env,
251              type="point2point",
252              end_effector_frame="r_hand_base_link",
253              hand_key_point=get_point_in_env(planner.env,point_name="
                     grasp_point_base_right_hand"), # Grasp point on the right hand
254              object_key_point=get_point_in_env(planner.env,type_name="can",obj_id
                     =0), # Center point of the can
255          )
256      )
257      # Add a Constraint to align the pinky-to-index axis of the right hand with the
             world z-axis
258      constraints.append(
259          Constraint(
260              env=planner.env,
261              type="parallel",
262              end_effector_frame="r_hand_base_link",
263              hand_axis=get_axis_in_env(planner.env,axis_name="right_ring_2_index"),
                     # The direction from the pinky finger to the index finger
264              object_axis=np.array([0,0,1]), # World frame. Or the specific axis of
                     the object, such as `get_axis_in_env(planner.env, axis_name="z",
                     obj_type="can", obj_id=0)`.
265          )
266      )
267      # Generate the target pose for the end-effector and move the right hand to the
             target pose
268      _, target_effector_pose = planner.generate_end_effector_pose(constraints,
             hand_name="right")
269      planner.move_to_pose_with_screw(target_effector_pose,"right",attach_obj=False)
270
271      # Close the right hand to grasp the can
272      planner.hand_grasp("right",grasp_object="can",obj_id=0)
273  ```
274  step 2: Lift the can by 0.1m while keeping its pose unchanged
275  ```python
276      constraints=[]
277      # Add a point-to-point constraint to lift the can by 0.1m
278      constraints.append(
279          Constraint(
```

```
280              env=planner.env,
281              type="point2point",
282              end_effector_frame="r_hand_base_link",
283              hand_key_point=get_point_in_env(planner.env,type_name="can",obj_id=0),
                   # can center point now
284              object_key_point=get_point_in_env(planner.env,type_name="can",obj_id
                   =0)+np.array([0,0,0.1]), # can center point after being lifted by
                   0.1m
285          )
286      )
287      # Add a parallel constraint to keep the x-axis of the can unchanged
288      constraints.append(
289          Constraint(
290              env=planner.env,
291              type="parallel",
292              end_effector_frame="r_hand_base_link",
293              hand_axis=get_axis_in_env(planner.env,axis_name="x",obj_type="can",
                   obj_id=0), # The x-axis of the can
294              object_axis=get_axis_in_env(planner.env,axis_name="x",obj_type="can",
                   obj_id=0), # Keep the x-axis unchanged
295          )
296      )
297      # Add a parallel constraint to keep the y-axis of the can unchanged
298      constraints.append(
299          Constraint(
300              env=planner.env,
301              type="parallel",
302              end_effector_frame="r_hand_base_link",
303              hand_axis=get_axis_in_env(planner.env,axis_name="y",obj_type="can",
                   obj_id=0), # The y-axis of the can
304              object_axis=get_axis_in_env(planner.env,axis_name="y",obj_type="can",
                   obj_id=0), # Keep the y-axis unchanged
305          )
306      )
307      # Generate the target pose for the end-effector and move the right hand to the
             target pose
308      _, target_effector_pose = planner.generate_end_effector_pose(constraints,
             hand_name="right")
309      planner.move_to_pose_with_screw(target_effector_pose,"right",attach_obj=True,
             object_name="can",object_id=0)
310
311      # Open the right hand to release the can
312      planner.open_hand("right")
313  ```
314  step3: Move the right hand back to its initial pose
315  ```python
316      # Move the right hand back to its initial pose
317      planner.move_to_pose_with_screw(planner.right_hand_init_pose, "right",
             attach_obj=False)
318  ```
319
320  2. Based on step 1 of Example 1, place the can into the bowl after grasping it.
321
322  step 2: Move the can above the bowl and ensure the palm is facing downward.
323  ```python
324      constraints=[]
325      # Add a point-to-point constraint to lift the can by 0.1m
326      constraints.append(
327          Constraint(
328              env=planner.env,
329              type="point2point",
330              end_effector_frame="r_hand_base_link",
331              hand_key_point=get_point_in_env(planner.env,type_name="can",obj_id=0),
                   # can center point now
332              object_key_point=get_point_in_env(planner.env,type_name="bowl",obj_id
                   =0)+np.array([0,0,0.1]), # can center point over the bowl
333          )
334      )
335      # Palm facing down
336      constraints.append(
337          Constraint(
338              env=planner.env,
339              type="parallel",
340              end_effector_frame="r_hand_base_link",
341              hand_axis=get_axis_in_env(planner.env,axis_name="right_grasp_axis"), #
                    The back of the palm points toward the front of the palm.
342              object_axis=np.array([0,0,-1]), # World frame,
343          )
```

```
344        )

346        # Generate the target pose for the end-effector and move the right hand to the
                target pose
347        _, target_effector_pose = planner.generate_end_effector_pose(constraints,
               hand_name="right")
348        planner.move_to_pose_with_screw(target_effector_pose,"right",attach_obj=True,
               object_name="can",object_id=0)

350        # Open the right hand to release the can
351        planner.open_hand("right")
352  ```
353  step3: Move the right hand back to its initial pose
354  ```python
355        # Move the right hand back to its initial pose
356        planner.move_to_pose_with_screw(planner.right_hand_init_pose, "right",
               attach_obj=False)
357  ```

359  ======
360  Notes:
361  - `type_id` represents different models of the same type of object, while `obj_id`
          indicates the order in which objects of the same type are added.
362  - The left hand's operable range is  x  [-0.42, -0.19],  y  [-0.07, 0.36]; the
          right hand's range is  x  [-0.42, -0.19],  y  [-0.36, 0.07]. It must be
          ensured that the hand moves within this range. For actions such as grasping,
          pinching, or placing an object, it must be ensured that they are all within
          the operational range of the corresponding hand.
363  - The object's bounding box (bbox) represents its dimensions in its own coordinate
          system (axis-aligned when the orientation quaternion is q =[1,0,0,0]). For
          example, a can with bbox: {"min": [-0.08,-0.08,-0.02], "max":
          [0.08,0.08,0.02]} has lengths of 0.16 units along X/Y-axes and 0.04 units
          along the Z-axis.
364  - Please place each step in a separate code block, enclosed within ```python ```
          tags. The number of steps should correspond exactly to the number of ```
          python ``` blocks in your response, with no extra or missing blocks.
365  - Variables in different code blocks are not shared, meaning variables from
          previous code blocks cannot be used in subsequent ones.
366  - When placing an object, release it slightly above the target position.
367  - To avoid collisions, consider splitting a move action into multiple moves. For
          example, when placing a can into a bowl, first move the can to a height above
          the bowl, then lower it to just above the bowl before releasing it. However,
          ensure the height is neither too high nor too low to avoid inverse
          kinematics issues or collisions. Typically, an additional height of 0.03 cm
          is added.
368  - When planning, please take collision issues into account. For example, if the
          height of an object relative to the table is z=0.03<0.08, do not use the
          right hand to grasp from right to left, as it will cause a collision between
          the hand and the table. Similarly, when placing the object somewhere, it is
          acceptable to release it slightly above the target position.
369  - If 'Executed Action Code' is empty, it means no code has been executed. If not
          empty, generate only the subsequent actions to be executed.
370  - If 'Prohibited Next Actions' is empty, there are no restrictions on the next
          action. If not empty, avoid executing any action identical to the listed ones
          in the first subsequent step.
371  - When performing operations such as moving an object to a specific location for
          placement, various factors should be fully considered. For example, when
          placing an object on a table, it is acceptable to suspend it slightly above
          the table.
372  - Before executing grasp or pinch actions, pre-grasp and pre-pinch actions must be
          performed, respectively. Before executing another grasp or pinch action, the
          corresponding pre-action must be performed again.
373  - When calling `Constraint`, parameter names must be explicitly specified using
          the `name=value` format.
374  - Each line of code must have at least one level of indentation.
375  - "right_hand_init_pose" and "left_hand_init_pose" are non-collision poses, which
          can avoid collisions with other hands or objects.
376  ======
377  Please generate the code to continue executing the task 'TASK_DESCRIPTION'.
378  TASK_NOTE
```

## E.6    Sample Demonstration Code for Block Stack Hard Generated by Applying MCTS

```
1  from humanoidgen.motion_planning.h1_2.solution.generated.solver_env import *
2  def step(planner:HumanoidMotionPlanner):
3      # Step 1: Right hand picks cube1 and places it at the stacking base position
4      planner.hand_pre_pinch("right")
5
6      constraints = [
7          Constraint(
8              env=planner.env,
9              type="point2point",
10             end_effector_frame="r_hand_base_link",
11             hand_key_point=get_point_in_env(planner.env, point_name="
                   pinch_point_base_right_hand"),
12             object_key_point=get_point_in_env(planner.env, type_name="cube",
                   obj_id=1)
13         ),
14         Constraint(
15             env=planner.env,
16             type="parallel",
17             end_effector_frame="r_hand_base_link",
18             hand_axis=get_axis_in_env(planner.env, axis_name="right_pinch_axis"),
19             object_axis=np.array([0, 0, -1])
20         )
21     ]
22     _, target_pose = planner.generate_end_effector_pose(constraints, "right")
23     planner.move_to_pose_with_screw(target_pose, "right", attach_obj=False)
24
25     planner.hand_pinch("right", pinch_object="cube", obj_id=1)
26
27     constraints = [
28         Constraint(
29             env=planner.env,
30             type="point2point",
31             end_effector_frame="r_hand_base_link",
32             hand_key_point=get_point_in_env(planner.env, type_name="cube", obj_id
                   =1),
33             object_key_point=np.array([-0.3, 0.0, 0.05])
34         ),
35         Constraint(
36             env=planner.env,
37             type="parallel",
38             end_effector_frame="r_hand_base_link",
39             hand_axis=get_axis_in_env(planner.env, axis_name="right_pinch_axis"),
40             object_axis=np.array([0, 0, -1])
41         )
42     ]
43     _, target_pose = planner.generate_end_effector_pose(constraints, "right")
44     planner.move_to_pose_with_screw(target_pose, "right", attach_obj=True,
           object_name="cube", object_id=1)
45
46     planner.open_hand("right")
47
48     planner.move_to_pose_with_screw(planner.right_hand_init_pose, "right",
           attach_obj=False)
49
50     # Step 2: Left hand picks cube0 and stacks it on cube1
51     planner.hand_pre_pinch("left")
52
53     constraints = [
54         Constraint(
55             env=planner.env,
56             type="point2point",
57             end_effector_frame="l_hand_base_link",
58             hand_key_point=get_point_in_env(planner.env, point_name="
                   pinch_point_base_left_hand"),
59             object_key_point=get_point_in_env(planner.env, type_name="cube",
                   obj_id=0)
60         ),
61         Constraint(
62             env=planner.env,
63             type="parallel",
64             end_effector_frame="l_hand_base_link",
65             hand_axis=get_axis_in_env(planner.env, axis_name="left_pinch_axis"),
66             object_axis=np.array([0, 0, -1])
67         )
68     ]
```

```python
69      _, target_pose = planner.generate_end_effector_pose(constraints, "left")
70      planner.move_to_pose_with_screw(target_pose, "left", attach_obj=False)
71
72      planner.hand_pinch("left", pinch_object="cube", obj_id=0)
73
74      constraints = [
75          Constraint(
76              env=planner.env,
77              type="point2point",
78              end_effector_frame="l_hand_base_link",
79              hand_key_point=get_point_in_env(planner.env, type_name="cube", obj_id
                  =0),
80              object_key_point=np.array([-0.3, 0.0, 0.09])
81          ),
82          Constraint(
83              env=planner.env,
84              type="parallel",
85              end_effector_frame="l_hand_base_link",
86              hand_axis=get_axis_in_env(planner.env, axis_name="left_pinch_axis"),
87              object_axis=np.array([0, 0, -1])
88          )
89      ]
90      _, target_pose = planner.generate_end_effector_pose(constraints, "left")
91      planner.move_to_pose_with_screw(target_pose, "left", attach_obj=True,
              object_name="cube", object_id=0)
92
93      planner.open_hand("left")
94
95      planner.move_to_pose_with_screw(planner.left_hand_init_pose, "left",
              attach_obj=False)
96
97      # Step 3: Right hand picks cube2 and completes the stack
98      planner.hand_pre_pinch("right")
99
100     constraints = [
101         Constraint(
102             env=planner.env,
103             type="point2point",
104             end_effector_frame="r_hand_base_link",
105             hand_key_point=get_point_in_env(planner.env, point_name="
                  pinch_point_base_right_hand"),
106             object_key_point=get_point_in_env(planner.env, type_name="cube",
                  obj_id=2)
107         ),
108         Constraint(
109             env=planner.env,
110             type="parallel",
111             end_effector_frame="r_hand_base_link",
112             hand_axis=get_axis_in_env(planner.env, axis_name="right_pinch_axis"),
113             object_axis=np.array([0, 0, -1])
114         )
115     ]
116     _, target_pose = planner.generate_end_effector_pose(constraints, "right")
117     planner.move_to_pose_with_screw(target_pose, "right", attach_obj=False)
118
119     planner.hand_pinch("right", pinch_object="cube", obj_id=2)
120
121     constraints = [
122         Constraint(
123             env=planner.env,
124             type="point2point",
125             end_effector_frame="r_hand_base_link",
126             hand_key_point=get_point_in_env(planner.env, type_name="cube", obj_id
                  =2),
127             object_key_point=np.array([-0.3, 0.0, 0.13])
128         ),
129         Constraint(
130             env=planner.env,
131             type="parallel",
132             end_effector_frame="r_hand_base_link",
133             hand_axis=get_axis_in_env(planner.env, axis_name="right_pinch_axis"),
134             object_axis=np.array([0, 0, -1])
135         )
136     ]
137     _, target_pose = planner.generate_end_effector_pose(constraints, "right")
138     planner.move_to_pose_with_screw(target_pose, "right", attach_obj=True,
              object_name="cube", object_id=2)
139     planner.open_hand("right")
```

# F Broader Impacts

This work advances the field of robotic manipulation by enabling automated task creation and demonstration collection for bimanual dexterous systems, which can significantly reduce the cost and complexity of data acquisition in humanoid robotics. Traditional methods often rely on manual demonstration design, expert supervision, or extensive environment resets—processes that are time-consuming, labor-intensive, and difficult to scale. By contrast, our framework automates these processes through an LLM-driven planning framework that generates semantically meaningful and physically feasible tasks without human intervention. This not only simplifies the pipeline from task specification to execution but also ensures consistency and diversity in the collected demonstrations.

The proposed framework has the potential to accelerate research on autonomous humanoid agents, particularly in domains such as assistive robotics, disaster response, and intelligent automation, where bimanual coordination and fine-grained hand manipulation are critical. For instance, in assistive robotics, precise bimanual operations, such as opening a pill bottle or pouring liquid from one container to another, require both high-level task reasoning and low-level motion control. Similarly, in disaster response scenarios, robots may need to manipulate tools, open doors, or handle irregularly shaped objects in constrained environments. These tasks demand robust and adaptive manipulation capabilities that go beyond simple pick-and-place actions. Our framework enables the generation of such complex interaction sequences, allowing researchers to train and evaluate policies on long-horizon, multi-step tasks at ease.

