# OpenReview forum: "HumanoidGen: Data Generation for Bimanual Dexterous Manipulation via LLM Reasoning"
_NeurIPS.cc/2025/Conference — NeurIPS 2025 poster_

### Official Review · Reviewer_uN4c · 2025-06-28

**Clarity:** 3
**Significance:** 3
**Originality:** 2
**Rating:** 4
**Confidence:** 3

**Summary:**

This paper introduces HumanoidGen, an automated framework for generating high-quality demonstration data for humanoid robots performing bimanual dexterous manipulation tasks. Unlike existing robotics datasets that primarily focus on single-arm manipulation, HumanoidGen specifically addresses the complexities of coordinating dual arms with dexterous hands. The framework employs three key components: (1) detailed spatial annotations for both assets and dexterous hands based on atomic operations, (2) an LLM-based planning system that generates chains of actionable spatial constraints for arm movements, and (3) a Monte Carlo Tree Search (MCTS) enhancement to improve LLM reasoning for complex long-horizon tasks. The authors develop a comprehensive benchmark called HGen-Bench containing 20 diverse manipulation tasks performed by a Unitree H1-2 humanoid robot with Inspire hands in simulation. Experiments demonstrate that HumanoidGen achieves superior success rates compared to existing methods, especially for long-horizon tasks with complex collision scenarios. The authors also show that the collected demonstration data effectively scales with 2D and 3D diffusion policies, confirming the quality and utility of the generated datasets.

**Questions:**

To what extent is your LLM-based approach truly automated? Does the pipeline require human filtering, validation, or intervention at critical stages, such as spatial annotation creation, constraint generation, or demonstration selection? If human oversight is needed, could you elaborate on these touchpoints and discuss how they might impact the scalability of your method for larger object libraries or more diverse task domains? Additionally, have you explored methods to further automate these potentially manual processes?

**Ethical Concerns:**

["NO or VERY MINOR ethics concerns only"]

**Final Justification:**

The authors' rebuttal has addressed several of my concerns; therefore, I will maintain my original score.

**Limitations:**

yes

**Quality:**

3

**Strengths And Weaknesses:**

Strengths:
- Effective system for data generation: The paper addresses the significant lack of high-quality bimanual dexterous manipulation data, which is crucial for advancing humanoid robotics.
- Comprehensive spatial annotation system: The detailed framework for annotating key points and axes for both assets and hands provides a solid foundation for LLM-based planning.
- Innovative constraint-based formulation: By framing manipulation as a sequence of spatial constraints rather than pre-defined functions, the approach enables more flexible and generalizable task execution.

Weaknesses:
- Reliance on extensive annotations: The approach requires detailed spatial annotations for both assets and hand operations, which could limit scalability to new objects and environments.
- Limited real-world validation: All experiments are conducted in simulation, leaving questions about how well the approach would transfer to real-world humanoid robots with physical constraints.
- Task complexity limitations: Despite improvements, the system still struggles with highly complex bimanual long-horizon tasks (e.g., HandOver and Storage), achieving lower success rates compared to simpler tasks.

---

> ### Author Rebuttal · Authors · 2025-07-31
>
> We thank the reviewer for the encouraging feedback and valuable comments.
>
> **Weaknesses**
>
> **W1:**
>
> **The scalability to new objects**: To evaluate the scalability of our framework across a diverse set of manipulable assets, we perform supplementary annotation experiments on five object categories: two rigid types of "cup" (with and without handle), "bowl", and two articulated objects, namely "laptop" and "drawer". For transferring annotations across similar objects, we employ the Stable Diffusion-based method introduced in the paper.
>
> | Evaluation Metric                                  | Cup (Handle) |  Cup (Edge)   |     Bowl      |    Laptop     |    Drawer    |
> | :------------------------------------------------- | :----------: | :-----------: | :-----------: | :-----------: | :----------: |
> | **Manual Annotation Time (s)**                     |     4.00     |     4.00      |     4.00      |     6.00      |     6.00     |
> | **Automatic Annotation Transfer Time (s)**         |     0.72     |     0.72      |     0.79      |     0.81      |     0.80     |
> | **Automatic Annotation Transfer Success Rate (%)** | 90.0 (18/20) | 100.0 (20/20) | 100.0 (20/20) | 100.0 (20/20) | 90.0 (18/20) |
>
> For a new object, manually annotating an autonomous action point along with its axis takes only 4 seconds for rigid objects and 6 seconds for articulated objects, thanks to the SAPIEN visualization interface and our annotation-assistive scripts. For automatic annotation transfer, the objects listed in the table are randomly selected from the SAPIEN asset library. The success rates are all above 90%, and the transfer time is under one second, which ensures the scalability of our approach to large-scale asset migration.
>
> Besides, the Stable Diffusion-based method is just one of the approaches we use to simplify the annotation process. Many existing works have explored affordance extraction and structured object axis annotation, such as RoboBrain(CVPR 2025), and CAPNET(CVPR 2025). These methods can be seamlessly integrated into our framework to support zero-shot asset annotation and enable large-scale asset expansion.
>
> **Scalability to New Scenes**: Our annotation process is object-centric. Once atomic actions are annotated for an object, they can be applied across different scenes involving that object. At the room level, our framework leverages the rich asset information in RoboCasa to directly generalize tasks to 120 scenes, significantly increasing scene diversity. At the table level, our framework integrates constraint-optimization-based target pose solving, dynamic active collision handling, and MCTS-enhanced reasoning. These techniques enable robust adaptation to task scenarios involving object randomization, cluttered environments, and diverse execution strategies.
>
> **W2:**
>
> Real-World Experiments: We thank the reviewer for pointing out the lack of real-world experiments in our manuscript. In response to the reviewer's suggestion, we have conducted real-robot experiments in both the data generation pipeline and the sim-to-real stages.
>
> We conduct experiments following 5 steps:
> 1）**Real-World Setup and Task Design.** We built a real scene corresponding to the simulation environment, which includes a Unitree H1_2 robot equipped with the Instant-Lab dexterous hand, a white table, and the task-related objects. Three tasks are designed for this data collection experiment: "bimanual bottle grasping", "opening a laptop", and "placing a block".
> 2) **Real2Sim Assets.** For task-related objects, we prepared both real objects and their corresponding simulation digital assets.
> Specifically, we obtained the digital assets of standard objects such as Coke cans and Sprite bottles from online asset repositories. For objects whose digital assets can hardly be found, such as the specific plate we used, we generate their digital assets by leveraging Hunyuan3D with fine-grained post-processing applied to ensure their consistency in geometric and appearance.
> 3) **Real2Sim Pose Estimation.** Using the existing digital assets, we estimate the objects' current pose in the camera coordinate system with FoundationPose(CVPR 2024). The 6D pose is then transformed into the world coordinate system, allowing the digital asset to be placed at the same pose in our simulation's world coordinate.
> 4) **Trajectory Collection.** We use LLM reasoning and the trajectory generation part of our framework to automatically generate execution trajectories in the simulation, and execute them on the real robot to successfully collect real-world data. We collected 25 trajectories for each task mentioned in 1) to train DP, trained for 300 iterations.
> 5) **Sim2Real Policy Transfer.** To investigate the effectiveness of simulation data in improving real-world robot policy training, we augment the real-world dataset collected above with additional simulation data. We aim to explore how simulation data contribute to the performance of trained robot policies in real-world experiments. For each aforementioned task, we evaluate the policy performance under the setting of training with 10 real-world trajectories plus N simulated trajectories. The results are as follows:
>
> | Evaluation Metric                 | Dual Bottles Pick | Close Laptop              | Notes                     |
> | --------------------------------- | ----------------- | ------------------------- | ------------------------- |
> | **Scene Randomization Range**     | 6 cm x 6 cm, ±15° | 6 cm x 6 cm, +10° to +20° | Position (cm) and Yaw (°) |
> | **Data Collection Success Rate**  | 85%               | 90%                       |                           |
> | **Data Collection Time**          | 16                | 13                        | Seconds per episode       |
> | **DP Success Rate(5Real)**        | 0%(0/20)          | 20%(4/20)                 | N=20 trials per condition |
> | **DP Success Rate(25Real)**       | 60%(12/20)        | **70%(14/20)**            | N=20 trials per condition |
> | **DP Success Rate(100Sim+5Real)** | **65%(13/20)**    | **70%(14/20)**            | N=20 trials per condition |
>
> As shown above, our approach consistently achieves over 85% success rate in real-world data collection, with each trajectory completed in just about 15 seconds. Furthermore, the simulation data produced by our framework significantly enhances real-world execution performance.
>
> **W3:**
>
> For complex tasks, which typically involve more execution steps and more intricate bimanual coordination, reasoning becomes more challenging for LLMs, leading to a significant drop in success rate compared to simpler tasks. Despite this challenge, our framework still maintains a success rate above 50%, showing its capability in handling bimanual complex and long-horizon tasks. Besides, this planning process can be further enhanced by MCTS to improve both the reasoning success rate and the diversity of execution plans. Furthermore, our collision avoidance mechanisms enable the framework to excel in executing tasks with complex collision scenarios. This is demonstrated by our significantly higher success rate in tasks involving complex collision scenarios compared to RoboTwin, as detailed in our manuscript.
>
>
> **Questions**
>
> **Q1:**
>
> We answer the questions as follows:
> 1) Our approach is automated with object annotations and the necessary atomic operations provided.
> 2) We list manual parts of our work as follows: On one hand, the atomic operations in our framework are currently manually predefined to ensure stable performance across diverse dexterous manipulation processes. On the other hand, for annotation transfer, our framework currently requires a separate manual annotation for one object per category to serve as a reference. Besides, we note that once these initial conditions are met, our framework enables fully automatic data collection, eliminating the need for manual effort in constraint generation or demonstration selection.
> 3) In terms of the manual labor involved in annotation transfer, it currently represents the minimal necessary effort required to scale our framework to a larger object library. We believe this level of manual input is highly efficient compared to fully manual annotation and significantly enhances the scalability of our framework. Moreover, we note that our framework is compatible with methods for automatic object annotation, such as affordance prediction and dexterous grasping pose generation, which could further reduce human intervention in the annotation process.
>
> **Q2:**
>
> We note some possible approaches for further automating the whole process:
> 1) Regarding the predefinition of atomic operations, we notice the recent advancements in research on dexterous hand manipulation pose generation. For example, DexGraspAnything(CVPR 2024) and DexVLG generate grasping and manipulation poses for dexterous hands based on 3D and/or 2D perception. Since 2D and 3D information can be provided accurately and reliably in simulation, we believe integrating these methods into our framework could further enhance the data collection process by automatically providing diverse and feasible end-effector pose candidates, reducing the need for manual definition and expanding the range of supported manipulation operations.
> 2) For object annotation, our framework supports the integration of additional methods to further automate the annotation process. These methods include affordance point and axis prediction approaches, such as CAPNET(CVPR 2025) and RoboBrain(CVPR 2025), which can provide the necessary annotations to enable the successful execution of our framework.
> In summary, our framework has high scalability, enabling the integration of additional methods to improve the level of automation in the processes mentioned above.

---

> > ### Comment · Reviewer_uN4c · 2025-08-05
> >
> > The authors' rebuttal has addressed several of my concerns; therefore, I will maintain my original score.

---

> > > ### Author Response · Authors · 2025-08-05
> > > **Response and Appreciation for Review Feedback**
> > >
> > > Dear Reviewer,
> > >
> > > We are delighted that our rebuttal has successfully addressed your concerns. We deeply value your support of our work and are thankful for the insightful feedback you provided during the review process.
> > >
> > > Best regards,
> > >
> > > The Authors

---

### Official Review · Reviewer_A6rG · 2025-06-29

**Clarity:** 3
**Significance:** 4
**Originality:** 4
**Rating:** 5
**Confidence:** 3

**Summary:**

This paper presents HumanoidGen, an automated framework for generating diverse bimanual dexterous manipulation tasks and collecting demonstrations for humanoid robots. The framework leverages atomic operations, extensive spatial annotations, and LLM-based task planners to generate chains of actionable spatial constraints, which are then solved by off-the-shelf optimizers. To handle long-horizon tasks and incomplete asset annotation, the authors introduce a MCTS variant to enhance reasoning. The proposed HGen-Bench benchmark covers 20 tasks with randomized scene configurations, and validates the generated data with 2D and 3D policies.

**Questions:**

- What hardware was used (GPU/CPU)? Approximately how much time does it take for asset annotation, demonstration generation (including LLM inference and trajectory planning), and overall data collection? Are there any observed computational bottlenecks?
- How do you determine if two assets are similar enough for annotation migration? Do you have accuracy or failure statistics for the migration process, and what are typical failure cases?
- Have you observed any LLM hallucinations in collision ignore list generation or planning with excessively large MCTS trees? How do you manage such complexity in the search process?

**Ethical Concerns:**

["NO or VERY MINOR ethics concerns only"]

**Final Justification:**

Overall, the paper is of high quality, and I am maintaining my score of 5.

**Limitations:**

yes

**Paper Formatting Concerns:**

The paper follows the formatting requirement.

**Quality:**

4

**Strengths And Weaknesses:**

**Strengths:**

- Automated and Scalable Pipeline: The system is highly automated, from asset annotation to scene/task/code generation and demonstration collection, which greatly reduces the labelling and handcrafted programming workload by human.
- Novel Method in Asset Annontation: The paper innovatively leverages Stable Diffusion encoders for automatic cross-asset annotation, minimizing the manual effort required for large-scale asset libraries.
- Strong LLM Integration: Planning is formulated at a constraint-chain code level and solved by an LLM-augmented pipeline, including an MCTS variant to enhance reasoning and self-correction for long-horizon tasks.
- Comprehensive Benchmark: The authors design a broad and diverse set of tasks (HGen-Bench) with clear evaluation metrics.

**Weaknesses:**

- Simulation-based Setup: Most experiments and policy learning are performed in simulation. The robustness and scalability to the real robot has not been validated yet.
- Limited Discussion of Auto-annotation: Although the Stable Diffusion approach is well motivated, the manuscript does not quantify annotation accuracy or failure rates for point-axes migration compared to manual annotation, also it lacks discussion for potential matching failure due to dissimilar assets.
- LLM Latency and Computational Load: The approach relies on multiple LLM calls and optimizer for path planning; the actual computational resource and bottleneck are not discussed.
- Potential Risk of Hallucination: The LLM is responsible for dynamically generating the collision ignore list, but the paper does not discuss potential errors caused by LLM hallucination, which could lead to physically invalid collisions and noisy data.

---

> ### Author Rebuttal · Authors · 2025-07-31
>
> Thank you for your insightful comments. We will explain your concerns point by point.
>
> **Weaknesses**
>
> **W1:**
>
> Real-World Experiments: We thank the reviewer for pointing out the lack of real-world experiments in our manuscript. In response to the reviewer's suggestion, we have conducted real-robot experiments in both the data generation pipeline and the sim-to-real stages.
>
> We conduct experiments following 5 steps:
>
> 1. **Real-World Setup and Task Design.** We built a real scene corresponding to the simulation environment, which includes a Unitree H1_2 robot equipped with the Instant-Lab dexterous hand, a white table, and the task-related objects. Three tasks are designed for this data collection experiment: "bimanual bottle grasping", "opening a laptop", and "placing a block".
>
> 2. **Real2Sim Assets.** For task-related objects, we prepared both real objects and their corresponding simulation digital assets.
> Specifically, we obtained the digital assets of standard objects such as Coke cans and Sprite bottles from online asset repositories. For objects whose digital assets can hardly be found, such as the specific plate we used, we generate their digital assets by leveraging Hunyuan3D with fine-grained post-processing applied to ensure their consistency in geometric and appearance.
>
> 3. **Real2Sim Pose Estimation.** Using the existing digital assets, we estimate the objects' current pose in the camera coordinate system with FoundationPose. The 6D pose is then transformed into the world coordinate system, allowing the digital asset to be placed at the same pose in our simulation's world coordinate.
>
> 4. **Trajectory Collection.** We use LLM reasoning and the trajectory generation part of our framework to automatically generate execution trajectories in the simulation, and execute them on the real robot to successfully collect real-world data. We collected 25 trajectories for each task mentioned in 1) to train DP, trained for 300 iterations.
>
> 5. **Sim2Real Policy Transfer.** To investigate the effectiveness of simulation data in improving real-world robot policy training, we augment the real-world dataset collected above with additional simulation data. We aim to explore how simulation data contribute to the performance of trained robot policies in real-world experiments. For each aforementioned task, we evaluate the policy performance under the setting of training with 10 real-world trajectories plus N simulated trajectories. The results are as follows:
>
> |Evaluation Metric              | Dual Bottles Pick | Close Laptop| Notes|
> |-|-|-|-|
> | **Scene Randomization Range** |  6 cm x 6 cm, ±15°  | 6 cm x 6 cm, +10° to +20°	   |Position (cm) and Yaw (°)|
> | **Data Collection Success Rate**  |  85% | 90%  |
> | **Data Collection Time**  |  16 |  13 |Seconds per episode
> | **DP Success Rate(5Real)** |  0%(0/20)  | 20%(4/20)  | N=20 trials per condition
> | **DP Success Rate(25Real)** | 60%(12/20)  | **70%(14/20)** |N=20 trials per condition
> | **DP Success Rate(100Sim+5Real)** | **65%(13/20)**  | **70%(14/20)**|N=20 trials per condition
>
> As shown above, our approach consistently achieves over 85% success in real-world data collection, with each trajectory completed in just about 15 seconds. Furthermore, the simulation data produced by our framework significantly enhances real-world execution performance.
>
>
> **W2:**
>
> We conducted a quantitative analysis of asset transferability for three types of objects: "cup", "laptop", and "drawer":
>
> | Evaluation Metric             | Cup (Handle) | Cup (Edge) | Bowl  | Laptop | Drawer |
> | :--------------------------- | :----------: | :--------: | :---: | :----: | :----: |
> | **Manual Annotation Time (s)** | 4.00         | 4.00       | 4.00  | 6.00   | 6.00   |
> | **Automated Migration Time (s)** | 0.72  | 0.72 | 0.79  | 0.81   | 0.80   |
> | **Migration Success Rate (%)** | 90.0 (18/20) | 100.0 (20/20) | 100.0 (20/20) | 100.0 (20/20) | 90.0 (18/20) |
>
> As shown above, the time for automated annotation migration is significantly shorter than manual annotation, demonstrating its human-intervention-free advantage. This is especially evident when annotating complex articulated objects, whose operation points and axes often do not align with the object's geometric center or axes, making manual annotation non-intuitive and difficult. In contrast, automated annotation transfer does not face this issue.
>
> However, transfer is not always guaranteed to succeed. On the one hand, there are inherent limitations in the transfer model itself, such as potential point deviations—particularly when the asset is very small or far from the camera. On the other hand, even within the same asset category, the same action may apply to different locations. For example, cups without handles are typically grasped around the center, whereas cups with handles are usually grasped at the handle. Therefore, it is necessary to perform the transfer separately for these two types of cups.
>
>
>
> **W3:**
>
> We conducted a quantitative analysis of time and resource consumption across different stages for 20 tasks. The statistics are as follows:
>
> |                               | Manual annotation | Automatic annotation | Scene Generation | Script Generate | Script Execution With Data Collection
> |-------------------------------|------|------|------|------|------|
> | Time | 4-6s  |  1.17s | 12.64s | 18.31s | 14.4s
> |Resource Cost | Manual|Auto|Auto LLM(avarage 2958Tokens)|Auto LLM(avarage 3745Tokens)|Auto Motion Planner(0.53s)
>
>
> The above table summarizes data on a per-task basis. However, tasks can vary significantly depending on the number of execution steps involved. On average, a single motion planning step takes approximately 0.53s, and the input to the code generation model typically contains around 3745 tokens per call.
>
>
> **W4:**
>
> We agree with the reviewer that LLMs can produce potential errors due to hallucination. We observed this problem in the first experiment of our paper, which led to the failure cases there. We believe this problem can be addressed by our MCTS-enhanced LLM reasoning approach. Specifically, in this approach, we handle such errors by storing relevant failure information at the corresponding node in the search tree where the error occurred. When MCTS revisits that node during future exploration, this information is provided to the LLM as part of the prompt. As shown in our second experiment in the paper, this approach allows the LLM to proactively avoid repeating similar mistakes, which can help it avoid such hallucinations. Besides, our framework executes LLM-generated plans in simulation to collect trajectories once the automatically generated task conditions are satisfied. This helps avoid potential issues related to data quality and experimental safety that may arise from LLM hallucinations.
>
>
> **Questions**
>
> **Q1:**
>
> The GPU and CPU used in our experiments are Nvidia RTX 4090 and Intel Core i9-14900K, respectively. The details of time consumption in our framework are mentioned in the "weakness" section. As for computational bottlenecks, we observed that such bottlenecks exist in GPU and RAM resources when parallelizing multiple simulation environments for training data collection.
>
>
> **Q2:**
>
> As mentioned in the "Weaknesses" section, transferable assets must belong to the same category and share identical geometric features; otherwise, the transfer may fail or introduce significant errors.
> Typical examples include cups with handles versus those without, or drawers with round handles versus horizontal handles, as discussed in the "weakness" section.
>
> **Q3:**
>
> Referring to the "Weaknesses" section, we do encounter hallucination issues when using LLMs. To manage this complexity, we record the specific cause of any execution error induced by hallucination, such as syntax errors, runtime failures, or task failures, at the corresponding tree node where truncation occurs. This mechanism helps prevent the recurrence of the same hallucination during subsequent planning exploration.

---

### Official Review · Reviewer_mhYR · 2025-06-30

**Clarity:** 2
**Significance:** 1
**Originality:** 2
**Rating:** 3
**Confidence:** 5

**Summary:**

This paper presents HumanoidGen, a framework for automatic data generation in bimanual dexterous manipulation, leveraging LLM-based planning, spatial annotation, and MCTS-enhanced reasoning. The system annotates both assets and robot hands with spatial keypoints and axes, and uses LLMs to generate code-like constraint chains representing manipulation plans. These are then converted into executable demonstrations through trajectory optimization. To handle insufficient annotations or long-horizon tasks, the authors introduce an MCTS-based introspective mechanism that improves LLM reasoning robustness. Based on the proposed framework, a benchmark, HGen-Bench, is constructed with 20 tasks, and the generated demonstrations are used to train diffusion policies.

**Questions:**

1. Could the authors more clearly and precisely define the intended scope and applicability of the proposed framework?
2. Can the authors discuss whether their framework is applicable to more complex, contact-rich manipulation tasks beyond the current benchmark?
3. Can the authors elaborate on the scalability potential of their method, especially in terms of annotation cost and generalization to diverse scenes and objects?

**Ethical Concerns:**

["NO or VERY MINOR ethics concerns only"]

**Limitations:**

yes

**Paper Formatting Concerns:**

-

**Quality:**

2

**Strengths And Weaknesses:**

The main strengths of this paper lies in the design of a complete and automated framework for generating bimanual manipulation demonstrations in simulation. The framework integrates spatial annotation, LLM-based constraint planning, and MCTS-enhanced reasoning to handle long-horizon and complex tasks. Its effectiveness is supported by empirical comparisons with RoboTwin, where HumanoidGen achieves significantly higher success rates in demonstration generation, highlighting the practical value of the proposed method.
However, there are several weaknesses in the paper that raise some concerns for me.
* The proposed data generation framework depends heavily on manually designed spatial annotations for both assets and dexterous hands. While the annotations are carefully constructed and detailed, this level of manual specification may not scale well to more diverse or real-world object sets, especially those with complex geometry or articulation. Moreover, the paper does not provide sufficient empirical evidence to support the scalability of this framework. This raises concerns about whether the approach can scale up effectively beyond the curated simulation environments used in the current benchmark.
* While the paper is framed in the context of humanoid robotics, its focus is exclusively on bimanual dexterous hand manipulation. However, "humanoid" typically encompasses broader challenges such as locomotion, mobile manipulation, and whole-body control. Given that the proposed framework and benchmark do not address these aspects, the use of “HumanoidGen” and “HGen-Bench” as naming choices feels overstated. I recommend the authors adjust their claims and more cautiously position their work to avoid overgeneralization.
* Even within the claimed setting of bimanual dexterous manipulation, the actual use of dual-arm coordination and dexterity is limited. From the videos provided by the authors on the project website, it is evident that although both arms are present in most tasks, only one arm is typically active at a time. Except for a few tasks like handover, most demonstrations could plausibly be completed by a single arm, and the generated solutions often resemble single-arm strategies. Moreover, while multi-fingered hands are used on the robot, their function is largely restricted to simple pick actions, which are not fundamentally different from standard two-finger grippers. Therefore, even within the scope of bimanual dexterous manipulation, the paper does not address the truly challenging aspects such as coordinated motion, contact-rich dual-arm manipulation, or in-hand dexterity, which casts doubt on the depth and significance of the contribution.

---

> ### Author Rebuttal · Authors · 2025-07-31
>
> Thank you for your valuable comments. We will explain your concerns point by point.
>
> **Weaknesses**
>
> **W1:**
>
> To address the concern about the scalability of our framework to a wider range of manipulable assets, we conduct further automatic annotation transferring experiments on 5 types of objects with the method mentioned in the paper.
>
> | Evaluation Metric             | Cup (Handle) | Cup (Edge) | Bowl  | Laptop | Drawer |
> | :--------------------------- | :----------: | :--------: | :---: | :----: | :----: |
> | **Manual Annotation Time (s)** | 4.00         | 4.00       | 4.00  | 6.00   | 6.00   |
> | **Automated Migration Time (s)** | 0.72  | 0.72 | 0.79  | 0.81   | 0.80   |
> | **Migration Success Rate (%)** | 90.0 (18/20) | 100.0 (20/20) | 100.0 (20/20) | 100.0 (20/20) | 90.0 (18/20) |
>
> Besides, we discuss integrating other works into our framework to effectively handle objects with complex geometry and articulation, as mentioned by the reviewer.
>
> On the one hand, for articulated objects, we note works that focus on predicting affordance points and articulation axes to guide manipulation. Some recent works, such as AffordanceLLM(CVPR 2024) and CAP-NET(CVPR 2025), demonstrate strong generalization capabilities across various object categories, including both numerous single rigid objects and articulated objects. We also note that the input for their prediction and the annotation transformation process, specifically object point cloud sampling and 2D-3D back-projection, can be performed in simulation without noise or error. We believe these methods can provide point and axis annotations on articulated objects, effectively addressing the reviewer's concern regarding articulated objects.
>
> On the other hand, for objects with complex geometry, which pose significant challenges in both annotation and manipulation, we note that this represents a distinct research area where current methods are still immature. Some works, such as DexGraspNet (CoRL 2024) and DexVLG (arXiv:2507.02747), explore generating dexterous hand grasping poses from 2D and/or 3D perception. Although these methods have certain generalization limitations, they show promise in supporting the manipulation of objects with complex geometry. Despite the immature state of research in this area, we note that our framework is compatible with such approaches, offering scalability to handle objects with complex geometry.
>
> **W2:**
>
> We apologize for any confusion the name "HumanoidGen" may have caused to the reviewer. We would like to clarify that we have explicitly stated that our work focuses on generating data for bimanual dexterous manipulation tasks in the "Data Generation for Bimanual Dexterous Manipulation" part of our title. Both the abstract and introduction sections center around bimanual dexterous manipulation tasks, and we haven't found any content in the main body of the paper that could easily lead readers to misunderstand our work as claiming support for locomotion or whole-body control.
>
> If the concern lies primarily in the naming, we are open to revising it by removing the "HumanoidGen:" part of our title to emphasize the scope of our work as reflected in the main title. Furthermore, we believe our framework also holds potential for future extension to coordinated manipulation involving the lower body.
>
> **W3:**
>
> We appreciate the reviewer's comments and concerns regarding the limitations of our work. To further demonstrate our framework's capability in handling tasks involving bimanual coordination and dexterous hand manipulation, we conducted experiments on five additional tasks. As shown in the table below, our framework achieves a high success rate across the five new tasks, demonstrating its scalability to new tasks.
>
> | Evaluation Metric          |Dual Lift Pot          | Rotate Safe Knob | Open Safe Door | Press Toaster | Blocks Stack Hard With Barrier |
> | :------------------------ | :---------------: | :--------------: | :------------: | :-----------: | :-----------------------------: |
> | **Scene Randomization**(Position, Yaw) | 12 cm × 6 cm, ±15° | 5 cm × 5 cm, ±15° | 5 cm × 5 cm, ±15° | 5 cm × 5 cm, ±15° | 6 cm × 8 cm, ±10° |
> | **Success Rate (%)**     | 93.00             | 91.09            | 95.24          | 87.72         | 73.00                           |
>
>
> We further address the reviewer's concern in detail below:
>
> 1) Coordinated Motion:
> Our framework's adaptability to bimanual coordination tasks is demonstrated by the experiments in our manuscript and the additional "lift pot" experiment. We note that our contribution to bimanual coordination lies in leveraging LLMs to reason about the execution sequence of both arms and to dynamically avoid collisions arising from inter-arm interference at the planning level. Furthermore, we apply task decomposition with MCTS to enable diverse strategies in bimanual manipulation. For instance, in the “dual_bottles_pick_hard” task, the LLM must not only infer the constraints for grasping and placing blocks, but also determine how the two arms should coordinate, including the order of execution, the stacking locations, and the choice of base block or position. These choices, such as stacking on an existing block or selecting an empty region, require the LLM to reason over multiple coordination strategies. Therefore, we believe our framework can effectively handle bimanual coordination tasks using the proposed planning methods.
>
> 2) Dexterous Hand Operation:
> Compared to parallel 2-finger grippers, the dexterous hand's superior structure provides greater flexibility in end-effector poses, but also introduces increased complexity in determining appropriate hand configurations during manipulation tasks. On the one hand, our work aims to strike a balance between flexibility and complexity by defining atomic operations, providing a simplified yet effective approach to enable automatic data collection for dexterous manipulation tasks. We also demonstrated our framework's capability to incorporate additional atomic dexterous hand operations through the additional tasks like   'Rotate Safe Knob' and 'Press Toaster' above. On the other hand, by adding more atomic operations and annotations, integrating additional annotation methods, and incorporating other in-hand manipulation policies as framework components, our framework can extend the current utilization methods of dexterous hands to new tasks and further expand and leverage the dexterity and flexibility of the dexterous hands.
>
> To summarize, our framework supports bimanual coordination through MCTS-enhanced LLM reasoning with collision management at the planning stage. Moreover, we provide a method to balance the flexibility advantage and the complexity inherent in dexterous hand manipulation, and leverage the framework's scalability to enable further utilization of dexterous hands.
>
> **Questions**
>
> **Q1:**
>
> We discussed this in the "Weaknesses" section. Overall, we define our framework to be a data collection framework for bimanual dexterous manipulation tasks, with the following key aspects to be noted:
>
> 1) We leverage LLMs to generate task scenes and plan scripts, obtaining manipulation trajectories through an optimization-based motion solver.
> 2) We leverage MCTS to enhance LLM reasoning for stable and diverse plan script generation.
> 3) We provide diverse atomic manipulation operations in our framework, including grasping, pinching, pressing, pulling, and rotating, as well as basic operations on articulated objects, supporting a variety of bimanual dexterous manipulation tasks.
> 4) Our framework's effectiveness is primarily demonstrated by our experiments with DP and DP3.
> 5) The scalability of our framework is demonstrated by the annotation transfer experiment and further enables integration with works on affordance prediction and dexterous grasping pose generation, allowing extension to a broader range of object types.
>
> To summarize, our framework focuses on bimanual dexterous manipulation tasks and demonstrates strong advantages in enabling automatic data collection for applications in this domain.
>
> **Q2:**
>
> Our framework is highly scalable, supporting extension to complex, contact-rich manipulation tasks. It leverages LLMs for script generation, enhanced by MCTS for higher planning success rate and diversity. Currently, it supports atomic operations such as grasping, pinching, pressing, and rotating, and can be expanded with new annotations, operations, and dexterous policies. In short, our framework efficiently scales to more complex tasks and richer atomic operations.
>
>
> **Q3:**
>
> We answer this question from the following perspectives:
>
> 1) Annotation scalability and generalization to objects: Our framework supports annotation transfer, enabling extension to diverse objects, as demonstrated in the additional experiments. Its modular design allows integration of new annotation methods (e.g., affordance prediction), atomic operations, and dexterous manipulation policies, facilitating generalization to more complex objects.
>
> 2) Scene scalability and generalization to new tasks: As stated in our manuscript, the generated task scripts can be executed directly across all 120 RoboCasa scenes without additional annotation effort. We also elaborate on our framework's adaptability in the response to Question 2, highlighting its capabilities at both the LLM reasoning level and the atomic operation level.
>
> To summarize, our framework demonstrates strong scalability and generalization, enabling effective adaptation to diverse objects, complex manipulations, and a wide range of unseen scenes and tasks through its flexible integration of high-level planning and low-level operation components.

---

> ### Author Response · Authors · 2025-08-05
> **Expectations for Addressing Reviewer Concerns**
>
> Dear Reviewer,
>
> We wanted to express our gratitude for your insightful feedback during the review process of our paper. We hope we have satisfactorily addressed all your concerns and demonstrated the improved quality of our work. If there are any additional points you'd like us to consider, please let us know. Your insights are invaluable to us, and we're eager to address any remaining issues.
>
> Thank you for your time and effort in reviewing our paper.
>
> Best regards,
>
> The authors

---

> ### Author Response · Authors · 2025-08-08
>
> Dear Reviewer mhYR, thanks for your thoughtful review. As the author-reviewer discussion period is near its end, we wonder if our rebuttal addresses your concerns. Please let us know if any further clarifications or discussions are needed!

---

### Official Review · Reviewer_NeLy · 2025-07-03

**Clarity:** 4
**Significance:** 2
**Originality:** 2
**Rating:** 4
**Confidence:** 5

**Summary:**

This paper introduces HumanoidGen, a framework to automatically generate tasks and demonstrations for bimanual dexterous manipulation, addressing a key data scarcity problem for humanoid robots. It uses an LLM planner, enhanced with Monte Carlo Tree Search (MCTS), to produce a sequence of spatial constraints that guide the robot's actions. The primary contribution is a scalable method for generating high-quality training data, validated by showing that diffusion policies trained on this data exhibit positive performance scaling.

**Questions:**

- Will any real-world experiments be added? Or what's the barrier that prevents deploying the data generation system to real robots?
- What's the difference and connection of the proposed method to the branch of augmentation-style data generation methods? What are the time and effort expenses of the two styles?

**Ethical Concerns:**

["NO or VERY MINOR ethics concerns only"]

**Limitations:**

No discussion on limitations in the main text. I read the limitations in the appendix, but I think there are quite a few unmentioned limitations, e.g., cluttered scenes, deformable objects such as cloth folding, and contact-rich tasks such as push-T.

**Quality:**

3

**Strengths And Weaknesses:**

## Strengths
- Presenting a very detailed data generation pipeline for bimanual dexterous manipulation.
- Also presenting a benchmark for humanoid manipulation.
- Achieving a higher data generation success rate than RoboTwin, and a credible data scaling-up trend in the DP & DP3 experiments.

## Weaknesses
- Lack of real-world experiments. I'm willing to see at least some small-scale verification, either by running the data generation pipeline in the real world or by sim-to-real transfer.
- While this paper focuses on zero-shot data generation, there is a line of recent work doing augmentation-style data generation, exemplified by MimicGen (CoRL'23), DexMimicGen (ICRA'25), and DemoGen (RSS'25). These methods require a small set of human-collected demos as a start, which is not required by HumanoidGen. However, the pipeline of HumanoidGen is quite long. If not all the components are well-automated for all the tasks, it still requires considerable human intervention. So, I am curious about the total time and effort cost of the zero-shot generation pipeline, and how it compares against the augmentation-style methods.
- Lack of discussions on limitations in the main text. See the comments in the limitation section.
- Lastly, a very small comment: I think the Primary Area of this paper should fall into Applications rather than Deep learning.

---

> ### Author Rebuttal · Authors · 2025-07-31
>
> Thanks for your careful and valuable comments. We will explain your concerns point by point.
>
> **Weaknesses**
>
> **W1:**
>
> In response to the reviewer's suggestion, we have conducted real-robot experiments in both the data generation pipeline and the sim-to-real stages.
>
> We conduct experiments following 5 steps:
> 1）**Real-World Setup and Task Design**. We built a real scene corresponding to the simulation environment. Two tasks are designed for experiments: "Dual Bottles Pick" and "Close Laptop".
> 2) **Real2Sim Assets**. For task-related objects, we prepared both real objects and their corresponding simulation digital assets.
> Specifically, we obtained the digital assets of standard objects such as Coke cans from online asset repositories. For objects whose digital assets can hardly be found, we generate their digital assets by leveraging Hunyuan3D with fine-grained post-processing applied to ensure their consistency in geometric and appearance.
> 3) **Real2Sim Pose Estimation**. Using the existing digital assets, we estimate the objects' current pose in the camera coordinate system with FoundationPose (CVPR 24). The 6D pose is then transformed into the world coordinate system, allowing the digital asset to be placed at the same pose in our simulation's world coordinate.
> 4) **Trajectory Collection**. We use LLM reasoning and the trajectory generation part of our framework to automatically generate execution trajectories in the simulation, and execute them on the real robot to successfully collect real-world data. We collected 25 trajectories for each task for 300 iterations.
> 5) **Sim2Real Policy Transfer**. To investigate the effectiveness of simulation data in improving real-world robot policy training, we augment the real-world dataset collected above with additional simulation data. We aim to explore how simulation data contribute to the performance of trained robot policies in real-world experiments. For each aforementioned task, we evaluate the policy performance under the setting of training with 5 real-world trajectories plus 100 simulated trajectories. The results are as follows:
>
> |Evaluation Metric              | Dual Bottles Pick | Close Laptop| Notes|
> |-|-|-|-|
> | **Scene Randomization Range** |  6 cm x 6 cm, ±15°  | 6 cm x 6 cm, +10° to +20°	   |Position (cm) and Yaw (°)|
> | **Data Collection Success Rate**  |  85% | 90%  |
> | **Data Collection Time**  |  16 |  13 |Seconds per episode
> | **DP Success Rate (5 Real)** |  0% (0/20)  | 20% (4/20)  | N=20 trials per condition
> | **DP Success Rate (25 Real)** | 60% (12/20)  | **70% (14/20)** |N=20 trials per condition
> | **DP Success Rate (100 Sim + 5 Real)** | **65% (13/20)**  | **70% (14/20)**|N=20 trials per condition
>
> As shown above, our approach consistently achieves over 85% success rate in real-world data collection, with each trajectory completed in just about 15 seconds. Furthermore, the simulation data produced by our framework significantly enhances real-world execution performance.
>
> **W2:**
>
> We note two differences between our approach and the augmentation-style methods:
> 1) The difference in **annotation types**. Our method requires annotating **operation points and axes on the assets**. These annotations correspond to different robot atomic operations, such as grasping and pinching. In contrast, MimicGen and DexMimicGen collect trajectories by teleoperation and annotate **different subtask segments in the trajectories** for further replay. Compared to their annotation type, which relies on collected task-related trajectories and is hardly generalizable to new tasks, our annotation type is task-agnostic and supports further task extension. By combining manual annotation and annotation transferring, our method allows building a large-scale library with annotated assets, supporting automatic script generation and data collection of diverse tasks.
>
> 2) The difference in **resource consumption**. The table below shows the time cost of our method at different stages:
>
> | Metric                        | Manual Annotation | Automatic Annotation | Scene Generation | Script Generation | Data Collection Execution |
> | :---------------------------- | :---------------: | :------------------: | :--------------: | :---------------: | :-----------------------: |
> | **Time (s)**            | 4-6               | 1.17                 | 12.64            | 18.31             | 14.40                     |
> | **Resource Cost**             | Human Operator    | Automated Process    | LLM (2,958 tokens avg.) | LLM (3,745 tokens avg.) | Motion Planner (0.53 s)   |
>
>
> We evaluated the time cost of our method versus augmentation-style methods under several scenarios:
> For new tasks with existing assets:
>    Our method can generate trajectories fully automatically, with no manual effort required. The fixed time cost per task before generating trajectory data is **30.95s**, which accounts for Scene Generation and Script Generation. The time to generate a single trajectory equals the execution time of one run, averaging **14.4s**.
>
> * DemoGen requires teleoperation to collect an initial trajectory and additional manual effort to crop and segment point clouds. It's fixed time cost averages **86s**. Generating an additional trajectory only takes **0.01s**, but this approach cannot provide data augmentation for vision-based policies and cannot guarantee that the generated data is always feasible.
> * DexMimicGen has an even longer preparation time, averaging **253s**. Besides data collection, manual segmentation into subtasks is required before trajectory generation. After generation, it still needs to execute trajectories to collect data, with each trajectory averaging **16s**.
>
> For new assets and new tasks:
>    Our method requires an additional **4–6s** of manual calibration. Nevertheless, it remains the most time-efficient overall. This efficiency stems from the fact that our approach does not rely on a pre-collected trajectory as the basis for generation; instead, it leverages annotated assets.
>
>
> **W3:**
>
> We appreciate the reviewer's suggestion on limitations. We will move the limitation section from the appendix to the main body of the paper and further analyze the limitations of our work. We provide a detailed discussion below in response to the limitations mentioned by the reviewer.
>
> - **Cluttered scenes**: Our framework can handle tasks with cluttered environments and distractor objects. This capability stems from the Active Collision Avoidance and Dynamic Collision Management mechanisms in our framework design. Specifically, the Active Collision Avoidance enables the LLM to plan steps that first move occluding objects aside before manipulating the target object. The Dynamic Collision Management, on the other hand, enables the LLM to determine which collision bodies to consider during path planning. These mechanisms together can effectively manage complex collision scenarios, such as those found in cluttered task environments. In addition, our MCTS reasoning enhancement enables the system to iteratively correct failed attempts and continue searching for a valid trajectory in case the LLM fails to generate a successful plan in one shot. To the best of our knowledge, such capabilities are not present in prior works such as robotwin(CVPR2024), gensim2, and DexMimicGen.
>
> - **Deformable objects**: Our framework currently doesn't support deformable object manipulation for two reasons: First, the Maniskill simulator lacks deformable object support. Second, our framework targets tasks with clear goals, making operations like flattening a crumpled shirt challenging due to the need for real-time feedback on fabric wrinkles and excessive LLM reasoning steps. However, extending our framework with more diverse atomic skills, such as training specific DP policies for subtasks like flattening, could enable support for these "goal-ambiguous" tasks, highlighting the system's extensibility.
>
> - **Contact-rich tasks**: Contact-rich tasks require rapid reactions to runtime feedback. For tasks like push-T and shirt-flattening, the need for fine-grained step-by-step corrections arises from their goal-ambiguous nature, where each action can lead to multiple outcomes without a fixed solution. This makes precise trajectory replication impossible in our current framework. However, our system can instead generate alternative strategies, such as directly grasping and placing the T-shaped object. To better handle such tasks in the future, a promising direction is to enrich the framework with additional model-based atomic operations.
>
> **W4:**
>
> Field of affiliation. We thank the reviewer for the suggestion regarding the appropriate field of affiliation. While our work is indeed related to applications, we believe it also appropriately falls within the scope of deep learning. Our work uses LLMs to generate expert demonstrations for bimanual dexterous manipulation, enhances reasoning with MCTS, and evaluates models like DP and DP3, all closely tied to deep learning methodologies. We appreciate the reviewer’s thoughtful suggestion.
>
> **Questions**
>
> **Q1:**
>
> We added real-robot experiments, which are discussed in the "Weaknesses" section. Regarding the difficulties in deploying the framework to real systems mentioned by the reviewer, we believe the main challenge lies in the sim-to-real gap of physics simulation and visual rendering. The former can be addressed by applying system identification to optimize the parameters of the actuators in the simulation. The latter can be addressed by modifying lighting and brightness settings in the simulator to match real-world conditions.
>
> **Q2:**
>
> Due to the character limit, we refer the reviewer to the discussion in the 'Weakness' section. In addition, our object annotations are reusable across multiple tasks, whereas augmentation-style data generation methods are typically limited to augmenting data only in terms of spatial position.

---

> > ### Comment · Reviewer_NeLy · 2025-08-05
> >
> > Thanks for the rebuttal. My concerns have been mostly addressed. In particular, I appreciate the real-world experiments and the detailed comparison against the related works. Thus, I will maintain my original positive recommendation.

---

> > > ### Author Response · Authors · 2025-08-06
> > > **Response and Appreciation for Review Feedback**
> > >
> > > Dear Reviewer,
> > >
> > > We are very pleased to hear that our rebuttal has addressed most of your concerns. It's encouraging to know that you appreciate our real-world experiments and the detailed comparison with related works. Your positive recommendation means a lot to us, and we sincerely thank you for your valuable feedback throughout the review process.
> > >
> > > Best regards,
> > >
> > >
> > > The Authors

---

### Note · Authors · 2025-08-11

Dear reviewers and AC,

We sincerely appreciate your valuable time and effort spent reviewing our submission. In our work, we present **HumanoidGen**, an automated task creation and demonstration collection framework for bimanual dexterous manipulation, designed to effectively overcome the current scarcity of expert data for dual-arm dexterous hands.  We also introduce **HGen-Bench**, a comprehensive benchmark for such manipulation tasks. As noted by the reviewers, our method features an **automated** (A6rG, NeLy, mhYR) and **scalable** (A6rG, NeLy) pipeline, enabling **effective** (uN4c, mhYR), **diverse** (A6rG, uN4c) and **high-quality** (NeLy, uN4c) data generation.  In the rebuttal, we have carefully addressed the reviewers' concerns and have added **experiments** as follows:

- We add experiments on real-world robots to demonstrate the effectiveness and scalability of our framework in real-world settings.

- We add experiments on framework cost analysis and a comparison with augmentation-based data generation methods to showcase the automation and low time cost of our approach.

- We add experiments on annotation transfer to demonstrate the scalability of our asset annotations.

- We add experiments on bimanual collaboration, obstacle-rich environments, and diverse manipulation strategies to highlight the scalability of our framework to a wide range of manipulation tasks and scenarios.

We provided further **clarifications** of the framework details and experiments as follows:

- We add an explanation clarifying the scope and applicability of our framework.

- We add an explanation of how our MCTS mechanism addresses the risk of hallucination.

- We add an explanation of the low-cost process for extending our framework to new assets.

- We add an explanation of our framework's scalability to more complex manipulation tasks.

Throughout the rebuttal and discussion phases, we addressed each of the reviewers' concerns individually, and no additional concerns emerged. We believe the supplementary experiments and clarifications further validate the novelty, practicality, and scalability of our approach.  We respectfully request the AC's further consideration of our submission.

We greatly appreciate your thoughtful evaluation.

Authors.

---

### Decision · Program_Chairs · 2025-09-17

**Decision:**

Accept (poster)

**Comment:**

The paper proposes HumanoidGen, a framework for automatically creating tasks and collecting demonstrations through atomic dexterous operations combined with LLM-based reasoning. A new benchmark is introduced and evaluated using both 2D and 3D diffusion policies.

The main concerns include:
1. Lack of real-world experiments (NeLy, A6rG, uN4c)
2. Requiring considerable human intervention and labels  (NeLy, mhYR, uN4c)
3. Missing comparison with augmentation-style methods (NeLy)
4. Overgeneralization of the scope and the actual use of dual-arm coordination and dexterity is limited (mhYR)
5. Lack computation cost description (A6rG)
6. Low performance on complex bimanual long-horizon tasks (uN4c)

The rebuttal adequately addressed most concerns. Three reviewers remained positive (1 Accept, 2 Borderline Accept), while the single negative reviewer (Borderline Reject) did not engage further. Given the overall consensus, the AC recommends acceptance, with the expectation that the authors incorporate the raised feedback into the final version.